# WHAT MATTERS IN TRANSFORMERS? NOT ALL ATTENTION IS NEEDED

## ABSTRACT

While scaling Transformer-based large language models (LLMs) has demonstrated promising performance across various tasks, it also introduces redundant architectures, posing efficiency challenges for real-world deployment. Despite some recognition of redundancy in LLMs, the variability of redundancy across different architectures in transformers, such as MLP and Attention layers, is under-explored. In this work, we investigate redundancy across different modules within Transformers, including Blocks, MLP, and Attention layers, using a similarity-based metric. Surprisingly, despite the critical role of attention layers in distinguishing transformers from other architectures, we found that a large portion of these layers exhibit excessively high similarity and can be pruned without degrading performance. For instance, Llama-2-70B achieved a 48.4% speedup with only a 2.4% performance drop by pruning half of the attention layers. Furthermore, by tracing model checkpoints throughout the training process, we observed that attention layer redundancy is inherent and consistent across training stages. Additionally, we further propose a method that jointly drops Attention and MLP layers, allowing us to more aggressively drop additional layers. For instance, when dropping 31 layers (Attention + MLP), Llama-2-13B still retains 90% of the performance on the MMLU task. Our work provides valuable insights for future network architecture design. The code will be released upon acceptance.

## 1 INTRODUCTION

Transformer-based large language models (LLMs) have significantly advanced AI research, achieving remarkable performance across various domains OpenAI (2024); Team (2024). However, scaling these models also introduces redundant architectures, namely overarching, leading to inefficiencies that complicate their real-world deployment Frantar et al. (2023); Sun et al. (2023), e.g., inflating deployment costs and resource demands. For instance, while the deployment cost of Llama-2-70B exceeds 128GB in FP16 precision—surpassing the capacity of a single A100 GPU—it can be reduced in depth without significantly impacting performance Gromov et al. (2024).

Although several previous works have been proposed to promote LLM efficiency via removing redundant parameters or architectures Frantar et al. (2023); Sun et al. (2023), these approaches often employ universal techniques that overlook the unique characteristics of transformer architectures. Specifically, transformer Vaswani (2017) architectures are composed of multiple stacked blocks, each containing an MLP layer and an Attention layer, which serve distinct functions and exhibit corresponding different levels of redundancy Wang et al. (2023); Shi et al. (2023) This motivates a deeper investigation into the specific redundancies within Transformers, with the goal of identifying and addressing the most critical modules.

In this work, we systematically explore the redundancy in three key Transformer components: *Block*, *MLP*, and *Attention*. Using a similarity-based metric Gromov et al. (2024); Men et al. (2024) , we evaluate the importance of each component and progressively drop those identified as redundant. We first apply a "*Block Drop*" approach but observe that removing entire blocks leads to significant performance degradation. This suggests a need for a more fine-grained strategy.

Upon further examination, we explore the separate pruning of MLP and Attention layers. Our findings reveal that while dropping MLP layers negatively affects performance, a substantial portion of Attention layers, i.e., the core of Transformer architectures which distinguish it from other

mainstream architectures (e.g., RWKV Peng et al. (2023) and Mamba Gu & Dao (2024)), can be pruned without degrading the model's performance. For instance, dropping 50% of the Attention layers in Llama-2-70B Touvron et al. (2023) results in comparable performance to the full model, indicating a high degree of redundancy in these layers.

Building on these insights, we propose a more flexible approach, "*Joint Layer Drop*", which targets both MLP layers and Attention layers. By combining the importance scores of these layers, we find that jointly dropping low-importance Attention and MLP layers yields better performance under high sparsity conditions compared to pruning only one type of layer.

Our work demonstrates that Attention layer redundancy is not only significant but also consistent across different training stages, indicating that this redundancy is an inherent property of Transformer architectures. These findings open the door to more efficient Transformer designs, reducing both memory (e.g., KV-Cache) and computational costs (e.g., inference speed), while maintaining performance.

In summary, our key contributions are as follows:

- Through an in-depth analysis of redundancy in three key Transformer components—*Block*, *MLP*, and *Attention*—we uncover a surprising level of redundancy within the *Attention*.

- We propose "Attention Drop", a simple yet effective algorithm for efficiently removing redundant Attention layers in a training-free manner. Additionally, we introduce "Joint Layer Drop", which further improves the performance at high dropping ratios by jointly targeting both Attention and MLP layers.

- Our extensive experiments demonstrate the effectiveness of dropping Attention, for instance, removing 50% of the attention layers in Llama-2-70B results in only a 2.4% performance reduction while achieving up to a 48.4% speedup.

- We further show that the attention layers remain consistently high redundancy throughout the training process, indicating it as an inherent property and providing valuable insights for future architecture design.

## 2 RELATED WORKS

**Large Language Models** Although Transformer-based Large Language Models (LLMs) have demonstrated promising performance across various tasks, their deployment costs still remain a significant challenge for practical usage Sun et al. (2023); Lin et al. (2024); Gromov et al. (2024). Transformer Vaswani (2017) models consist of multiple blocks, which include Attention layers and MLP layers. Attention layers compute the contextual information between input tokens with quadratic complexity concerning the input sequence length Li et al. (2020). KV-Cache Pope et al. (2022) mitigates the computational issue but results in excessive memory costs Zhang et al. (2023). MLP layers Liu et al. (2021); Mai et al. (2022) transform each token independently, using an up-projection followed by a down-projection, and contribute most of the model parameters. Recent works have revealed that not all blocks or layers are equally important Men et al. (2024); Chen et al. (2024), which urges us to reflect on the structured redundancy within LLMs and the potential design of more compact architectures.

**Model Compression** LLMs can be compressed to promote their efficiency in memory and computation. Quantization Frantar et al. (2023); Lin et al. (2024) and Pruning Sun et al. (2023); Frantar & Alistarh (2023) are the most widely used techniques to compress LLMs. Specifically, quantization transforms the data type into low-bit but remains potentially redundant architecture and parameters. Pruning can be categorized into unstructured pruning Kusupati et al. (2020); Sanh et al. (2020) and structured pruning Zhuang et al. (2020); Kwon et al. (2020). While unstructured pruning maintains better performance than structured pruning, it cannot be effectively applied to hardware, limiting its practical usage. Our methods, Block Drop and Layer Drop, focus on removing structured modules rather than fine-grained parameters, creating hardware-friendly efficient architectures while maintaining comparable performance. Additionally, Block Drop and Layer Drop are orthogonal to quantization, and their integration with quantization significantly enhances efficiency.

## 3 METHODOLOGY

In this section, we present the methodology for identifying and removing redundant modules in LLMs. We begin by introducing a similarity-based metric to assess redundancy across both Attention and MLP layers. Based on the insights gained from this analysis, we develop two targeted techniques, i.e., MLP Drop and Attention Drop, to efficiently eliminate redundant components while preserving model performance.

### 3.1 PRELIMINARIES

**Similarity-based Drop** To assess the redundancy of modules in LLMs, we employ a similarity-based metric that evaluates the importance of each module by measuring the similarity between its input and output Gromov et al. (2024). The underlying hypothesis is that redundant modules produce outputs that are similar to their inputs, implying minimal transformation. In contrast, important modules are expected to significantly alter their inputs and thus should be preserved. The similarity between the hidden states of the input $X$ and output $Y$ of a module is quantified using cosine similarity. The importance score $S$ of the module is computed as:

$$S = 1 - \text{Cosine}(X, Y). \tag{1}$$

Modules with higher cosine similarity exhibit lower importance scores, indicating redundancy. We identify and prune the modules with the lowest importance scores according to a predefined prunting ratio. A complete evaluation of the metric's effectiveness is provided in Appendix B.

**Block Drop** Transformer models are composed of stacked blocks, where each block shares a common architecture and can be viewed as a subnetwork. To reduce complexity, we first consider dropping entire blocks that are deemed unimportant.

As shown in Figure 2, Transformer blocks operate sequentially, with each block's output feeding into the next. To evaluate redundancy, we compute the similarity between the input and output of each block. For the $l$-th block, the importance score is calculated as:

$$S_B^l = 1 - \text{Cosine}(X_B^l, Y_B^l), \tag{2}$$

where $X_B^l$ and $Y_B^l$ denote the input and output of the $l$-th block, respectively. Since the similarity scores are computed locally, we can offload irrelevant modules to save memory. By iteratively computing the importance scores for each block from shallow to deep, we can identify and drop blocks with the lowest scores, thus saving memory and computational resources.

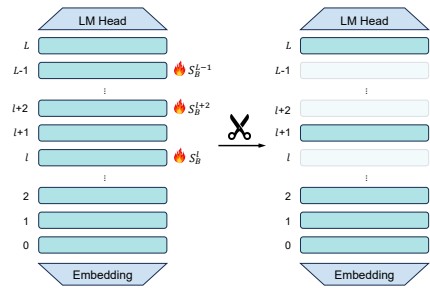

Figure 1: **Visualization of Block Drop**. where we use 🔥 to denote the blocks with high similarity scores. The dropped blocks are blurred.

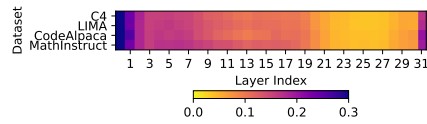

Figure 2: Importance scores of Blocks.

### 3.2 MOTIVATION

Block Drop is an aggressive technique that risks removing essential layers, as it overlooks the internal fine-grained architectures within each block. Given a transformer block consists of both an Attention layer and an MLP layer. These layers perform distinct functions, with the Attention layer facilitating contextual information flow between tokens and the MLP layer transforming the token representations. Given their distinct roles, we assess the redundancy of each layer separately by measuring the importance scores of Attention and MLP layers individually. Specifically, we leverage multiple calibration datasets to measure the importance scores, ranging from the pretraining dataset (e.g., C4 Raffel et al. (2020)) to instruction fine-tuning datasets (e.g., CodeAlpaca-20k[1], MathInstruct Yue et al. (2024) and LIMA Zhou et al. (2024)). Figure 2 and 3 illustrate the varying trend of importance scores for Attention layers compared to MLP layers across multiple datasets. This observation motivates us to consider the varying levels of redundancy between MLP and Attention

---

[1]https://huggingface.co/datasets/sahil2801/CodeAlpaca-20k

layers and to develop more fine-grained dropping techniques accordingly, namely, MLP Drop and Attention Drop.

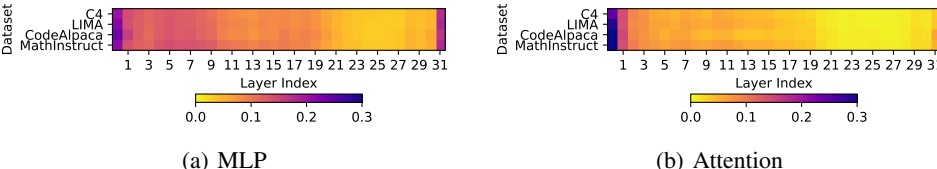

(a) MLP        (b) Attention

Figure 3: Importance scores for MLP and Attention layers, where we use various calibration datasets for a comprehensive analysis.

### 3.3 LAYER DROP.

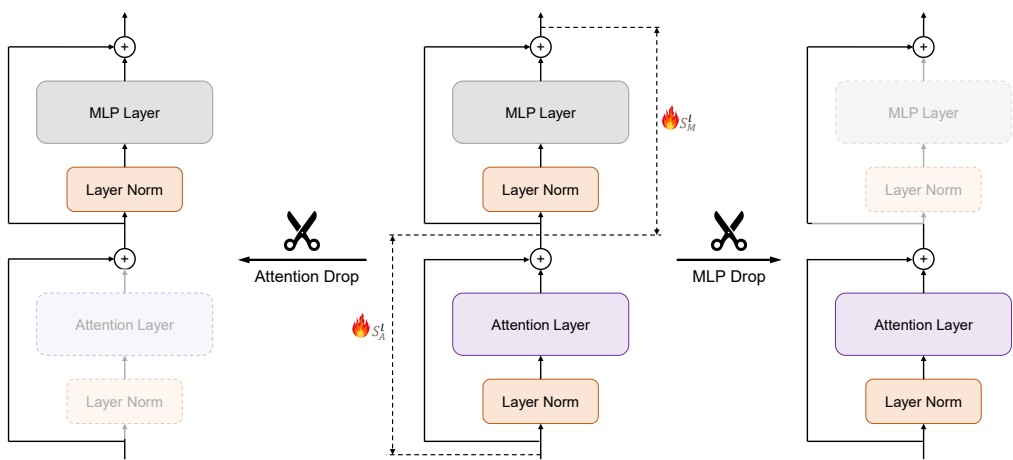

Figure 4: **Visualization of Layer Drop**, where we visualize dropping either MLP or Attention Layers. Given the residual connection, we take LayerNorm together with the corresponding layers. The dropped layers with high similarity scores 🔥 are blurred.

**MLP Drop** As illustrated in Figure 4, each MLP layer follows a LayerNorm operation and involves a residual connection, ensuring that part of the input is preserved in the final output. Given the input $\boldsymbol{X}_M^l$ of the LayerNorm before MLP at the $l$-th Block, the output $\boldsymbol{Y}_M^l$ can be formulated as:

$$\boldsymbol{Y}_M^l = \boldsymbol{X}_M^l + \text{MLP}(\text{LayerNorm}(\boldsymbol{X}_M^l)). \tag{3}$$

Since the output $\boldsymbol{Y}_M^l$ contains both the residual and the MLP transformation, evaluating similarity based solely on the MLP's output can be misleading. To address this, we consider the MLP layer and its associated LayerNorm as a single unit and compute their importance score as follows:

$$\boldsymbol{S}_M^l = 1 - \text{Cosine}(\boldsymbol{X}_M^l, \boldsymbol{Y}_M^l). \tag{4}$$

By treating these layers as a single entity, we ensure a more accurate measure of importance. MLP Drop removes both the unimportant MLP and associated LayerNorm layers. Further validation of this approach can be found in Appendix B.

**Attention Drop** Similarly, Attention layers also operate within a residual connection. The output of the $l$-th Attention layer is computed as:

$$\boldsymbol{Y}_A^l = \boldsymbol{X}_A^l + \text{Attention}(\text{LayerNorm}(\boldsymbol{X}_A^l)), \tag{5}$$

where $\boldsymbol{X}_A^l$ is the inputs of the corresponding LayerNorm layers and $\boldsymbol{Y}_A^l$ overall outputs that involves residual connections. Like MLP Drop, we assess both the Attention layer and its associated LayerNorm as a single unit. The importance score for the Attention layer is:

$$\boldsymbol{S}_A^l = 1 - \mathrm{Cosine}(\boldsymbol{X}_A^l, \boldsymbol{Y}_A^l). \tag{6}$$

Layer Drop, for both MLP and Attention layers, is performed in a one-shot manner, calculating importance scores once and removing redundant layers in a single step. This approach avoids the resource-intensive and time-consuming iterative pruning process. The effectiveness of this simple one-shot technique is evaluated in Appendix B.

**Implementation and Loading the Pruned Model**  After removing redundant layers, the pruned model can be easily loaded using existing libraries, such as Huggingface Transformers Wolf et al. (2020), with only minor adjustments to the model configuration. Additional implementation details are provided in Appendix A.

## 4 INVESTIGATION OF DROPPING DIFFERENT TARGET MODULES

In this section, we conduct a comprehensive investigation into the effects of dropping different target modules. To quantify the trade-off between performance degradation and speedup, we introduce a new metric, i.e., **Speedup Degradation Ratio (SDR)**, defined as:

$$\gamma = \frac{\Delta\mathrm{Avg.}}{\Delta\mathrm{Speedup}}, \tag{7}$$

where $\Delta$Avg. represents the percentage change in average performance across the evaluated tasks, and $\Delta$Speedup denotes the corresponding percentage of speedup achieved by each method. Therefore, $\gamma$ measures the amount of performance degradation incurred for each 1% increase in speedup. A lower $\gamma$ value indicates that the model achieves speedup with minimal performance loss, making it more efficient. In contrast, a higher $\gamma$ value suggests that the performance loss is substantial relative to the speedup gained, implying a less favorable trade-off.

Table 1 and Figure 5 summarize the results of dropping different target modules, such as Block, MLP, and Attention layers. Specifically, Table 1 shows the performance impact of dropping a fixed number of modules (e.g., 4 and 8 layers), while Figure 5 extends this analysis by evaluating a broader range of dropping ratios (0% to 100%).

**Block and MLP Drop: Significant Performance Degradation with Moderate Speedup**  Block Drop and MLP Drop both lead to notable performance declines across both models, despite achieving moderate speedups. For example, Dropping 4 blocks results in a 2.4% average performance decline (from 68.2 to 65.8) for Llama-2-13B, with a speedup of $1.11\times$, corresponding to a $\gamma$ of 0.22. However, dropping 8 blocks causes a 7.5% performance drop (down to 60.7), with only a modest speedup of $1.24\times$ and a higher $\gamma$ of 0.31. Similarly, MLP Drop exhibits a comparable trend, with a small decline at 4 layers (1.3%, $\gamma = 0.32$), but a much larger drop at 8 layers (6.3%, $\gamma = 0.79$). **These results suggest that while Block and MLP Drop provide moderate speedup, they do so at the cost of significant performance degradation, especially at higher drop ratios.**

**Attention Drop: Minimal Performance Impact with High Efficiency**  Surprisingly, despite the critical role of attention layers in Transformer architectures, dropping attention layers is highly effective. Both Llama-2-13B and Mistral-7B maintain over 99% of their original performance even after dropping 8 attention layers, as shown in Figure 5(c). For example, Attention Drop maintains near-baseline performance even after dropping 8 layers (69.8 vs. 70.3), with a speedup of $1.23\times$ and a low $\gamma$ of 0.02. Dropping 12 attention layers results in only a slight performance decline (67.3), with a significant speedup of $1.40\times$ and a $\gamma$ of 0.08. The superior performance of Attention Drop persists when compared to other compression techniques B. **These results demonstrate that attention layers are highly redundant, and their removal has minimal impact on model accuracy, making Attention Drop a highly efficient pruning strategy.**

Table 1: **Experimental Results of Dropping Different Modules**, where we drop the fixed number (e.g., 4 and 8) of modules on Llama-2-13B and Mistral-7B. Here, Block, MLP, and Attn are corresponding modules. Rows with averaged performance lower than 95% of the original performance are grayed.

| Llama-2-13B | | | | | | | | | | | |
|---|---|---|---|---|---|---|---|---|---|---|---|
| **Method** | ARC-C | BoolQ | HellaSwag | MMLU | OBQA | PIQA | RTE | WinoGrande | Avg. (↑) | SpeedUp (↑) | $\gamma$ (↓) |
| Baseline | 59.9 | 80.7 | 82.2 | 55.1 | 45.6 | 80.5 | 65.0 | 77.0 | 68.2 | 1.00× | – |
| Block-4 | 54.8 | 73.3 | 80.6 | 54.8 | 45.8 | 79.1 | 60.3 | 77.5 | 65.8 | 1.11× | 0.22 |
| Block-8 | 48.0 | 56.8 | 75.3 | 53.8 | 41.2 | 75.3 | 59.9 | 75.6 | 60.7 | 1.24× | 0.31 |
| MLP-4 | 54.9 | 76.1 | 80.4 | 54.8 | 45.4 | 79.5 | 66.4 | 77.3 | 66.9 | 1.04× | 0.32 |
| MLP-8 | 49.2 | 63.4 | 75.6 | 54.5 | 42.2 | 76.0 | 59.2 | 75.1 | 61.9 | 1.08× | 0.79 |
| Attn-4 | 58.8 | 80.4 | 82.0 | 54.7 | 46.2 | 80.5 | 67.9 | 77.2 | 68.5 | 1.05× | -0.05 |
| Attn-8 | 58.2 | 80.5 | 82.2 | 54.5 | 47.0 | 80.5 | 64.3 | 77.4 | 68.1 | 1.13× | 0.01 |
| Attn-16 | 56.4 | 79.2 | 81.9 | 48.2 | 47.4 | 79.5 | 59.9 | 76.2 | 66.1 | 1.29× | 0.07 |
| Attn-20 | 53.8 | 76.9 | 78.6 | 51.5 | 44.4 | 77.6 | 59.2 | 77.1 | 64.9 | 1.40× | 0.08 |
| **Mistral-7B** | | | | | | | | | | | |
| **Method** | ARC-C | BoolQ | HellaSwag | MMLU | OBQA | PIQA | RTE | WinoGrande | Avg. (↑) | SpeedUp (↑) | $\gamma$ (↓) |
| Baseline | 61.5 | 83.7 | 83.2 | 62.5 | 43.8 | 82.0 | 66.8 | 78.5 | 70.3 | 1.00× | – |
| Block-4 | 53.1 | 80.4 | 77.5 | 61.6 | 40.0 | 77.6 | 70.0 | 76.6 | 67.1 | 1.14× | 0.23 |
| Block-8 | 40.0 | 71.6 | 63.9 | 60.0 | 30.6 | 69.3 | 63.9 | 69.7 | 58.6 | 1.32× | 0.37 |
| MLP-4 | 53.2 | 80.3 | 77.7 | 61.7 | 40.0 | 77.6 | 67.5 | 77.3 | 66.9 | 1.03× | 1.13 |
| MLP-8 | 36.7 | 71.8 | 33.6 | 53.3 | 30.6 | 68.0 | 66.8 | 66.6 | 53.4 | 1.06× | 2.82 |
| Attn-4 | 61.0 | 83.5 | 82.9 | 62.5 | 44.6 | 82.0 | 64.6 | 78.0 | 69.9 | 1.10× | 0.04 |
| Attn-8 | 60.2 | 82.7 | 82.3 | 62.2 | 44.2 | 81.3 | 66.8 | 78.8 | 69.8 | 1.23× | 0.02 |
| Attn-12 | 57.2 | 76.8 | 80.2 | 59.4 | 41.8 | 79.1 | 66.1 | 77.7 | 67.3 | 1.40× | 0.08 |

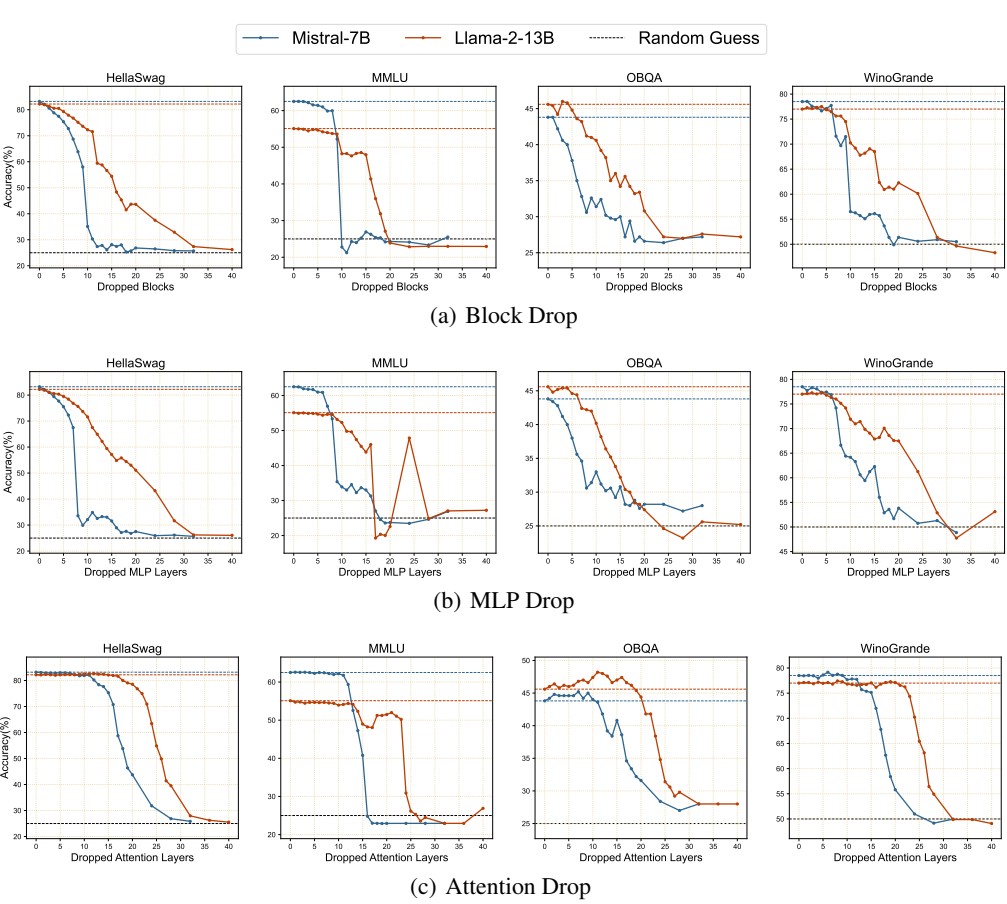

Figure 5: **Performance with respect to Dropping Ratios**. The solid lines represent the impact of dropping the $n$ modules with the lowest importance scores in Mistral-7B and Llama-2-13B, and the dotted lines represent the performances of the baseline and random guessing.

**Larger Models Show Consistent Robustness to Attention Drop** To verify the consistency of our findings on larger models, we take Llama-2-70B into consideration, since it also comes from the Llama family and larger model size. Specifically, we drop the modules with different dropping ratios ranging from 5% to 60% on Table 3.

Similar to the findings in smaller models, Llama-2-70B also showcases sensitivity to Block Drop and Layer Drop, where dropping only 10% to 20% of blocks or MLP layers leads to a significant performance drop.

In contrast, Attention Drop performs much better on Llama-2-70B. Specifically, when dropping 40 out of 80 attention layers, Llama-2-70B achieves a speedup of 1.48 and a $\gamma$ of 0.05. A similar trend is observed in Llama-3, as shown in Table 2. **This robustness indicates that larger models can also tolerate the removal of a significant proportion of Attention layers without degrading performance.**

The results clearly indicate that Attention Drop is the most efficient method for pruning, allowing for significant speedup with minimal impact on performance. In the following sections, we will examine the efficiency improvements achieved through Attention Drop and further investigate the layer-wise importance of attention layers to gain deeper insights into model architecture. For additional comparisons between Attention Drop and other compression methods, please refer to Appendix B.

Table 2: Experimental results on Llama-3, where Llama-3-8B and Llama-3-70B are included. Rows with averaged performance lower than 95% of the original performance are grayed.

| Method | HellaSwag | MMLU | OBQA | WinoGrande | Avg. (↑) | SpeedUp (↑) | γ (↓) |
|---|---|---|---|---|---|---|---|
| | | | Llama-3-8B | | | | |
| Baseline | 82.2 | 65.5 | 45.0 | 77.7 | 67.6 | 1.00× | – |
| Attn-4 | 81.6 | 65.1 | 44.8 | 78.2 | 67.4 | 1.07× | 0.03 |
| Attn-8 | 81.1 | 65.1 | 45.0 | 78.4 | 67.4 | 1.16× | 0.01 |
| Attn-12 | 79.4 | 63.9 | 42.2 | 77.8 | 65.8 | 1.26× | 0.07 |
| Attn-16 | 71.2 | 38.2 | 39.4 | 72.8 | 55.4 | 1.38× | 0.32 |
| Attn-20 | 42.2 | 23.0 | 30.6 | 58.7 | 38.6 | 1.52× | 0.56 |
| | | | Llama-3-70B | | | | |
| Baseline | 88.0 | 78.7 | 48.4 | 85.4 | 75.1 | 1.00× | – |
| Attn-4 | 87.9 | 78.7 | 49.0 | 85.2 | 75.2 | 1.04× | -0.03 |
| Attn-8 | 87.8 | 78.5 | 48.8 | 85.2 | 75.1 | 1.10× | 0.00 |
| Attn-16 | 87.8 | 78.7 | 48.6 | 84.9 | 75.0 | 1.17× | 0.01 |
| Attn-32 | 87.9 | 78.6 | 48.8 | 85.3 | 75.2 | 1.35× | 0.00 |
| Attn-40 | 85.2 | 77.1 | 48.0 | 82.8 | 73.3 | 1.43× | 0.00 |
| Attn-48 | 81.2 | 73.9 | 47.4 | 81.3 | 71.0 | 1.55× | 0.07 |

Table 3: **Block Drop and Layer Drop on Larger Models**, where we drop a series of numbers (from 4 to 48) of modules on Llama-2-70B. Rows with averaged performance lower than 95% of the original performance are grayed.

| | | | | Llama-2-70B | | | | | | | |
|---|---|---|---|---|---|---|---|---|---|---|---|
| Method | ARC-C | BoolQ | HellaSwag | MMLU | OBQA | PIQA | RTE | WinoGrande | Avg. (↑) | SpeedUp (↑) | γ (↓) |
| Baseline | 67.4 | 83.8 | 87.1 | 68.5 | 48.6 | 82.5 | 69.3 | 83.7 | 73.9 | 1.00× | – |
| Block-4 | 63.8 | 80.4 | 84.6 | 60.2 | 48.0 | 81.6 | 71.1 | 78.0 | 71.0 | 1.07× | 0.41 |
| Block-8 | 59.1 | 77.5 | 81.3 | 55.1 | 46.2 | 81.0 | 68.2 | 73.2 | 67.7 | 1.14× | 0.44 |
| Block-16 | 44.6 | 64.6 | 69.9 | 29.3 | 40.0 | 75.2 | 51.6 | 59.7 | 54.4 | 1.30× | 0.65 |
| Block-32 | 35.1 | 58.8 | 56.7 | 25.7 | 36.8 | 71.7 | 54.5 | 55.3 | 49.3 | 1.67× | 0.37 |
| MLP-4 | 65.4 | 84.0 | 86.1 | 68.7 | 46.6 | 82.9 | 68.2 | 83.4 | 73.2 | 1.04× | 0.18 |
| MLP-8 | 64.4 | 83.9 | 84.9 | 68.7 | 47.6 | 81.7 | 66.8 | 82.2 | 72.5 | 1.05× | 0.28 |
| MLP-16 | 57.5 | 53.6 | 81.6 | 69.1 | 46.0 | 79.2 | 58.8 | 81.7 | 65.9 | 1.08× | 1.00 |
| MLP-32 | 40.6 | 61.9 | 64.2 | 59.8 | 29.8 | 64.2 | 52.7 | 72.7 | 55.7 | 1.17× | 1.07 |
| Attn-4 | 67.2 | 84.0 | 87.0 | 68.6 | 48.8 | 82.5 | 69.3 | 83.3 | 73.8 | 1.06× | 0.02 |
| Attn-8 | 67.3 | 83.8 | 86.9 | 68.5 | 48.4 | 82.9 | 69.0 | 82.6 | 73.7 | 1.12× | 0.02 |
| Attn-16 | 67.8 | 83.9 | 87.2 | 68.5 | 49.0 | 83.0 | 68.2 | 82.8 | 73.8 | 1.21× | 0.00 |
| Attn-32 | 67.2 | 84.8 | 87.2 | 68.4 | 49.6 | 81.8 | 67.5 | 83.5 | 73.8 | 1.35× | 0.00 |
| Attn-40 | 63.7 | 82.8 | 84.4 | 66.2 | 46.8 | 80.1 | 66.8 | 81.3 | 71.5 | 1.48× | 0.05 |
| Attn-48 | 58.5 | 73.7 | 80.6 | 56.8 | 45.0 | 79.8 | 59.6 | 81.0 | 66.9 | 1.62× | 0.11 |

## 5 EFFICIENCY OF ATTENTION DROP

In this section, we evaluate the efficiency of Attention Drop in terms of both memory usage and inference speed. Specifically, we examine the reduction in memory overhead due to the key-value (KV) cache and measure the speed-up during the entire generation phase. The results demonstrate that Attention Drop provides substantial improvements in both efficiency metrics while maintaining high performance.

**KV-cache Memory Reduction** Given the auto-regressive nature of attention, where outputs are generated token by token, the KV-cache is used to store intermediate representations of input sequences. This cache helps accelerate inference by preventing redundant computations but comes with a significant memory cost, especially with longer sequence lengths or larger batch sizes. Our proposed Attention Drop method efficiently removes unimportant attention layers, reducing the corresponding KV-cache. Table 4 provides a comparison of 16-bit precision KV-cache memory usage before and after Attention Drop for various models, where we use 8 Nvidia RTX A6000

Ada GPUs for the 70B models and 4 Nvidia RTX A6000 Ada GPUs for other smaller models. As shown, Attention Drop results in substantial memory savings across all tested models. For instance, in Llama-2-13B, the KV-cache is reduced from 52GB to 26GB, a 50% reduction. This memory reduction is even more pronounced in larger models like Llama-2-70B, where the KV-cache decreases from 20GB to 10GB. Note the reported results are based on resource-constrained scenarios. In resource-sufficient cases, where larger batch sizes and longer sequence lengths can be applied, the memory usage savings from Attention Drop become even more significant. These reductions are beneficial for both memory-constrained and memory-sufficient environments, allowing for more efficient model deployment.

Table 4: **Comparison of KV-cache sizes before and after Attention Drop** across different models, with a sequence length of 2048. Since only Llama-2-13B does not use grouped-query attention, the KV-cache for each token is significantly larger compared to other models.

| Model | Batch Size | wo/Attn Drop | | w/Attn Drop | |
|---|---|---|---|---|---|
| | | Layers | KV-cache | Layers | KV-cache |
| Mistral-7B | 64 | 32 | 16GB | 20 | 10GB |
| Llama-2-13B | 32 | 40 | 52GB | 20 | 26GB |
| Llama-2-70B | 32 | 80 | 20GB | 40 | 10GB |
| Llama-3-8B | 64 | 32 | 16GB | 20 | 10GB |
| Llama-3-70B | 32 | 80 | 20GB | 40 | 10GB |

**Speed Measurement** We also evaluate the run-time speed improvements achieved through Attention Drop. The inference speed is measured throughout the entire generation process, starting from the input prompt to the generation of the final token. To ensure that the results accurately reflect the speed improvements, we follow two key principles in our setup: (1) all operations are performed on a single Nvidia RTX A6000 Ada GPU, avoiding any communication overhead caused by multi-GPU setups; and (2) we increase the batch sizes to maximize GPU utilization for each model. Specifically, for Llama-2-70B, we employ 4-bit quantization due to its large model size, while noting that Attention Drop is orthogonal to quantization shown in C. For Llama-2-13B and Mistral-7B, we use 16-bit precision. In terms of sequence length, we use an input sequence of 2048 tokens and autoregressively generate an additional 2048 tokens. This setup allows us to capture the full inference process, ensuring that both the prefill (initial processing of the input sequence) and the generation (token-by-token inference) stages are included in the speed measurements.

The speed-up ratios achieved through Attention Drop are presented in Tables 1, 2, and 3. Our results show that Attention Drop provides up to 40% speed-up while retaining more than 95% of the original model's performance. Additionally, as demonstrated in Table 1, the $\gamma$ values for Attention Drop are significantly lower than those for MLP Drop and Block Drop, especially at higher speed-up ratios. This indicates that Attention Drop achieves a more efficient trade-off between speed and performance, making it a superior method for model acceleration.

## 6 VISUALIZATION EXAMPLES OF LAYER IMPORTANCE

In this section, we visualize the importance scores and the corresponding dropping order of pretrained models. We then trace back through historical checkpoints to explore the dynamics of importance scores throughout the training process.

### 6.1 DEEPER MODULES WITH HIGHER REDUNDANCY

Based on Figure 2, 3 and 10, we observe that the deeper layers (excluding the last ones) often exhibit excessively low importance across Block, MLP, and Attention modules.

To further analyze the dropped modules, we visualize the dropped layers or blocks with different dropping ratios. Figure 6 visualizes the remaining and dropped layers/blocks as the number of dropped modules increases. Llama-2-13B and Mistral-7B exhibit similar patterns in Layer Drop and Block Drop: initially, both models tend to drop the deeper layers, followed by the shallower ones. These findings are consistent with Xu *et al.* Men et al. (2024), which suggests that deeper layers tend to be more redundant. Larger models (e.g., Llama-2-70B) also showcase a similar trend, which is shown in Appendix C.

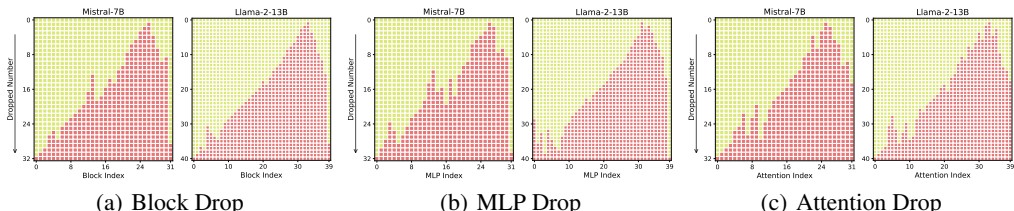

|                | (a) Block Drop | (b) MLP Drop | (c) Attention Drop |

Figure 6: **Visualization of Dropping Order for Block Drop and Layer Drop**. We visualize the remaining layers and blocks under different dropped numbers, where yellow areas represent the retained layers/blocks and red areas indicate the dropped portions.

## 6.2 CONSISTENT REDUNDANCY OF ATTENTION LAYERS THROUGHOUT TRAINING

Now that the deep layers exhibit high redundancy, to investigate how such a pattern is achieved, we revisit the historical checkpoints to track the dynamic changing of layer-wise importance scores.

Specifically, we use checkpoints released by MAP-Neo-7B Zhang et al. (2024), since it released continuous checkpoints during training stages. Figure 7 presents the importance scores of Blocks and Layers at different training stages, where Attention layers demonstrate consistently lower importance scores than MLP and Block at all training stages. While the importance scores for MLP layers and Blocks gradually increase as training progresses, the importance scores of Attention layers change much more slowly.

Given the consistently higher redundancy of attention layers throughout training, we believe that this pattern arises from the inherent properties of attention layers.

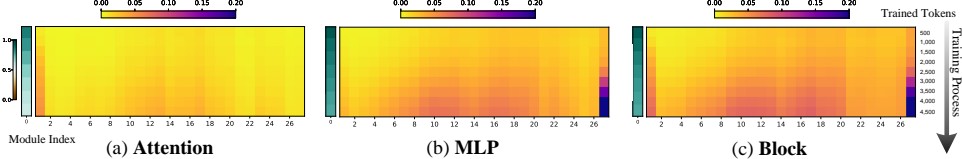

|  | (a) **Attention** | (b) **MLP** | (c) **Block** |

Figure 7: **Visualization of Importance Scores in Checkpoints during the Pre-training Process of MAP-Neo-7B**, where lighter areas represent low importance scores (i.e., high similarity scores). We present the entire Training Process (checkpoints for every 500B trained tokens). We independently visualize the importance scores at the module index 0, since they are significantly higher.

## 7 JOINT LAYER DROP FURTHER ENHANCES THE PERFORMANCE

While a significant proportion of attention layers exhibit high redundancy, our findings also show that some MLP layers have low importance. To further optimize model efficiency, we introduce Joint Layer Drop, which combines both Attention Drop and MLP Drop strategies. This approach leverages the redundancy in both attention and MLP layers to enhance the overall performance of the model.

**Methodology: Combining Attention and MLP Drop**   The Joint Layer Drop method is implemented by first calculating the importance scores for both attention layers ($\boldsymbol{S}_A^l$) and MLP layers ($\boldsymbol{S}_M^l$) individually. These scores are computed based on a similarity-based metric that identifies redundant layers, as previously discussed. Once the importance scores are obtained for each type of layer, we concatenate the scores into a single array: $\boldsymbol{S} = [\boldsymbol{S}_A^l, \boldsymbol{S}_M^l]$. From this combined set of importance scores, we drop the layers with the lowest values, regardless of whether they are attention or MLP layers. This joint approach allows us to remove the most redundant components from both layer types simultaneously, enhancing the model's efficiency while maintaining performance.

**Superior Performance with Joint Layer Drop**   As demonstrated in Figure 8, Joint Layer Drop consistently achieves better performance than either Attention Drop or MLP Drop alone. The process begins by exclusively dropping attention layers, which are typically more redundant than MLP layers. This continues until the number of dropped attention layers exceeds 14 for Mistral-7B and 18 for Llama-2-13B. As a result, in the initial stages of pruning, the performance of Joint Layer Drop overlaps with that of Attention Drop.

However, as the dropping ratio increases and the more redundant attention layers are pruned, MLP layers start to become the next most redundant components. At this point, Joint Layer Drop begins to remove MLP layers, leading to further reductions in redundant layers without significant performance loss, e.g., after dropping 31 layers (Attention + MLP), Llama-2-13B still retains 90% of the performance on the MMLU task.

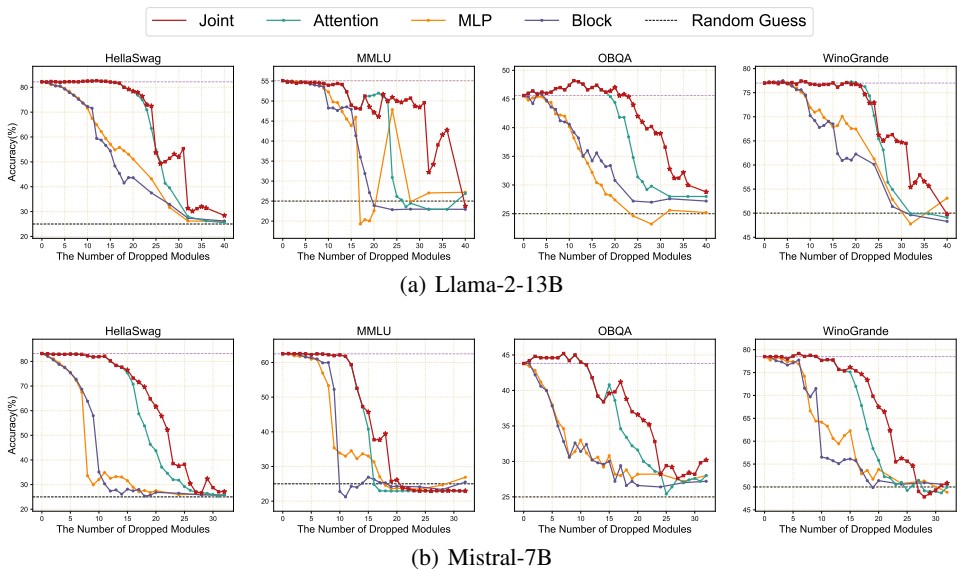

(a) Llama-2-13B

(b) Mistral-7B

Figure 8: **Accuracy Curves of Dropping Different Target Modules**, where we consider dropping single types of modules and Joint dropping (Attn + MLP). In the line of Joint Drop, ★ represents the step where the MLP is dropped, while the ■ represents the step where the Attention is dropped.

## 8 CONCLUSION AND DISCUSSIONS

**Insights for Future Network Architecture Design**   Despite the success of scaling up LLMs, our work provides valuable insights for scaling down models to achieve more efficient architectures. One key insight is the high redundancy in attention layers, particularly in deeper layers, which suggests that future works could reduce the number of attention layers without sacrificing performance, rather than maintaining parity with MLP layers. Moreover, unlike MLP layers, attention layers exhibit consistent redundancy as training progresses. This consistency may pose a bottleneck in training large models. To address this, future research could explore replacing attention layers with alternative mechanisms or develop new training techniques that capitalize on this redundancy to further enhance language model capacity.

**Limitations**   While our proposed dropping techniques improve efficiency in the models we evaluated, there are limitations. A key area for future work is testing the applicability of these techniques across a broader range of models, such as vision transformers and vision-language models. Furthermore, our methods focus on post-training dropping without involving retraining, which could potentially recover or even improve performance after pruning. Retraining these models could unlock even greater efficiency in more compact architectures.

**Conclusion**   In this work, we systematically revisited transformer architectures by investigating the effects of dropping three types of structures: Blocks, MLP layers, and Attention layers. Our findings reveal that attention layers display significant redundancy and can be removed in large proportions without compromising performance. To build on this, we introduced Joint Layer Drop, a method that further increases both dropping ratios and performance by targeting redundant layers across both MLP and Attention layers. This study empirically demonstrates the potential for creating more compact and efficient transformer models, providing valuable insights for future network design within the NLP community. By exploring structured redundancy, we open up new avenues for designing more efficient, scalable models that maintain high performance even under resource constraints.

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

# A    IMPLEMENTATION DETAILS

**Models**    We utilize Llama-2 Touvron et al. (2023) and Mistral Jiang et al. (2023) as the default models, given their competitive performance and wide usage. We also evaluated the completely open-source model MAP-Neo Zhang et al. (2024) to explore the redundancy variations in the modules of the entire pre-training phase. Additionally, we experimented with the newly released Llama-3 to verify the effectiveness of model dropping on the latest models.

**Datasets**    For the calibration dataset, we used the validation set of C4 dataset Raffel et al. (2019), with 256 samples and an input sequence length of 2,048, following the setup in Sun et al. (2023). The setting is well-supported by Appendix B. To evaluate model performance, we report normalized zero-shot or few-shot accuracy on the LM-harness benchmark, which includes multiple tasks: ARC-C Clark et al. (2018), BoolQ Clark et al. (2019), HellaSwag Zellers et al. (2019), MMLU Hendrycks et al. (2021), OBQA Mihaylov et al. (2018), PIQA Bisk et al. (2019), RTE Wang et al. (2019), and WinoGrande ai2 (2019). Please refer to Table 5 for detailed information. The evaluation code is based on EleutherAI LM Harness Gao et al. (2023).

Table 5: **Experimental settings for evaluation tasks.** "Norm" refers to the normalization performed with respect to the length of the input.

| Task | Number of few-shot | Metric |
|---|---|---|
| BoolQ | 0 | Accuracy |
| RTE | 0 | Accuracy |
| OBQA | 0 | Accuracy (Norm) |
| PIQA | 0 | Accuracy (Norm) |
| MMLU | 5 | Accuracy |
| WinoGrande | 5 | Accuracy |
| GSM8K | 5 | Exact Match |
| HellaSwag | 10 | Accuracy (Norm) |
| ARC-C | 25 | Accuracy (Norm) |

# B  ABLATION STUDIES

**One-Shot v.s. Iterative**   One-shot and iterative approaches are the two most common methods for model compression. In the one-shot approach, importance scores are computed once, and the model is pruned in a single step. In contrast, the iterative method computes importance scores and prunes the model incrementally over multiple iterations. In Figure 9, we empirically compare Iterative Dropping and One-Shot Dropping, where in Iterative Dropping, layers are removed one by one in each iteration.

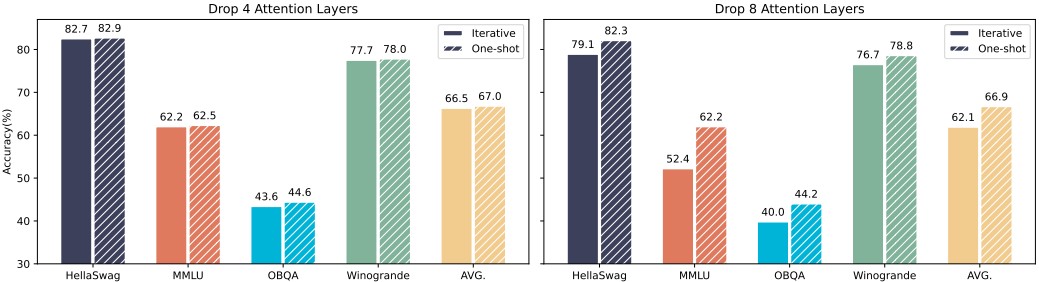

Figure 9: **Ablation Study on Dropping Strategies**, i.e., Iterative and One-Shot, where One-Shot Dropping achieves comparable performance with Iterative Dropping.

As shown in Figure 9, Iterative Dropping achieves performance that is merely comparable to One-Shot Dropping, without offering any significant enhancement. Given the simplicity and efficiency, One-Shot Dropping emerges as the superior choice.

Table 6: **Ablation Study on the Residual Connection**, where we report the average performance on MMLU, WinoGrande, HellaSwag, and OpenbookQA. $n$ denotes the number of dropped modules. The notation "w/ res" indicates the involvement of the residual connection, while "w/o" indicates dropping without considering it.

|       | Attn Drop |         | MLP Drop |         |
|-------|-----------|---------|----------|---------|
| $n$   | w/o res   | w/ res  | w/o res  | w/ res  |
| 4     | 39.4      | 65.0    | 31.2     | 64.5    |
| 8     | 37.7      | 65.3    | 31.1     | 61.9    |
| 12    | 36.8      | 65.4    | 31.1     | 55.6    |
| 16    | 32.2      | 63.4    | 30.8     | 49.9    |
| 20    | 32.0      | 62.9    | 30.9     | 42.1    |

**Residual Connection**   The involvement of the residual connection ensures a more accurate estimation by accounting for the overall inputs and outputs. To explore its impact on performance, we also consider dropping modules without involving the residual connection. In this case, the importance scores are measured solely by the inputs and outputs of the Attention or MLP layers. As shown in Table 6, the involvement of the residual connection is essential for Layer Drop.

**Calibration Datasets**   Figure 3 demonstrates the robustness of the importance scores across different datasets. In Figure 10, we further verify that the importance scores remain relatively stable across various modules of Mistral-7B as the sample size increases. This stability indicates that both Block Drop and Layer Drop maintain consistency regardless of the number of samples. Consequently, we confirm that using 256 samples is sufficient for computing similarity, which serves as the standard adopted for all our experiments.

**Dropping Metrics**   We selected the Reverse Order and Relative Magnitude metrics proposed by ShortGPT Men et al. (2024) and applied them to Attention Drop. Additionally, we considered the random dropping of attention layers. Our experiments were conducted using the Mistral-7B model, and the reported performance is averaged across five different random seeds. Notably, in Table 7, our metric, Cosine Similarity, consistently outperformed the others.

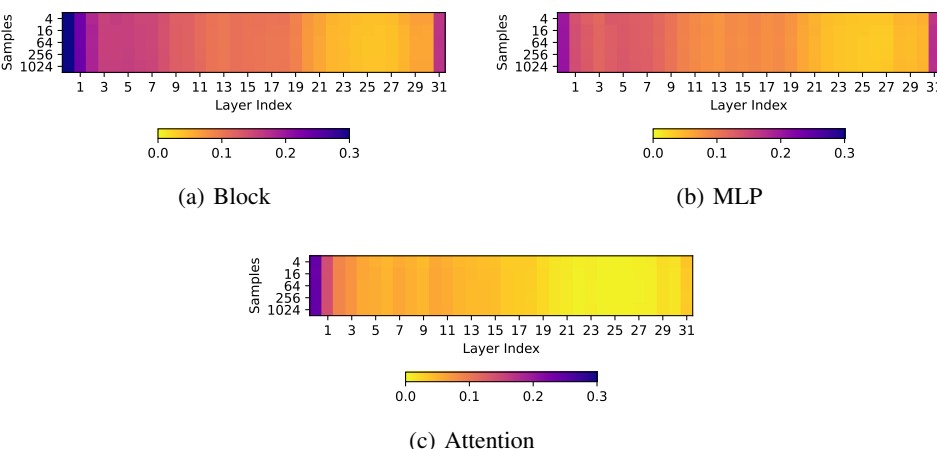

Figure 10: The impact of sample quantity on the importance scores of Block, MLP and Attention.

Table 7: Ablation study of Attention Drop across different metrics.

| Metric | Attn-4 | Attn-8 | Attn-12 |
|---|---|---|---|
| Random | 61.5 | 49.6 | 39.4 |
| Reverse Order | 66.9 | 66.9 | 61.5 |
| Relative Magnitude | 67.0 | 66.8 | 62.3 |
| Cosine Similarity | **67.0** | **66.9** | **64.8** |

**Comparison with other Compression Techniques**   We first compare our method with published sparse models pruned by Shortened LLaMA Kim et al. (2024) in Table 8. Specifically, we prune the original Vicuna-13B-v1.3 model Chiang et al. (2023) using our proposed Joint Drop technique to maintain the same parameter budget. While Shortened LLaMA involves post-compression retraining, training-free Joint Drop performs better on average performance. We also compare our approach with the mainstream pruning method Wanda Sun et al. (2023) in Table 9. Under the same parameter budget, our methods outperform Wanda with unstructured sparsity. Additionally, Wanda contributes to fine-grained sparsity, which is not hardware-friendly and has limited practical usage.

Table 8: Comparison with Shortened LLaMA Kim et al. (2024), where Joint Layer Drop achieves significantly higher speedup.

| Method | HellaSwag | MMLU | OBQA | Winogrande | Avg. (↑) | SpeedUp (↑) |
|---|---|---|---|---|---|---|
| Joint Layer Drop | 76.0 | 49.6 | 42.4 | 74.9 | 60.7 | 1.45 |
| Shortened LLaMA-PPL | 75.3 | 47.7 | 44.2 | 74.0 | 60.3 | 1.23 |
| Shortened LLaMA-Taylor | 76.8 | 47.0 | 42.4 | 76.3 | 60.6 | 1.22 |

Table 9: Comparison with Wanda Sun et al. (2023) under the same parameter budget. Taking performance into account, we apply unstructured sparsity for Wanda, while our proposed Attention Drop outperforms it in both performance and efficiency.

| Method | HellaSwag | MMLU | OBQA | Winogrande | Avg. (↑) | SpeedUp (↑) |
|---|---|---|---|---|---|---|
| Wanda | 82.3 | 54.8 | 45.2 | 77.6 | 65.0 | 1.00× |
| Attn-4 | 82.0 | 54.7 | 46.2 | 77.2 | 65.0 | 1.05× |
| Wanda | 82.4 | 54.7 | 45.8 | 77.6 | 65.1 | 1.00× |
| Attn-8 | 82.2 | 54.5 | 47.0 | 77.4 | 65.3 | 1.13× |
| Wanda | 82.4 | 54.8 | 46.2 | 77.4 | 65.2 | 1.00× |
| Attn-12 | 82.7 | 54.4 | 48.0 | 76.6 | 65.4 | 1.20× |

## C ADDITIONAL EXPERIMENTAL RESULTS

**Dropping Order on Larger Models** We present the dropping order of Block Drop and Layer Drop for the 70B Llama models in Figure 11. Similar to smaller models, larger models also tend to drop deeper layers first. While the dropping order of Blocks differs between Llama-2-70B and Llama-3-70B, we believe this is attributed to different training techniques, e.g., different numbers of training tokens.

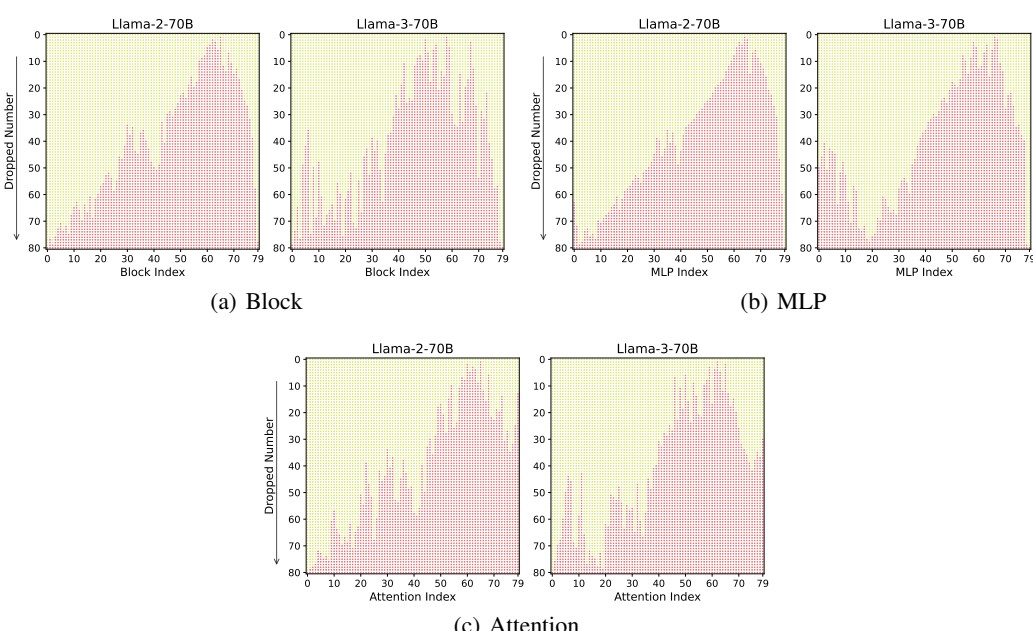

Figure 11: **Visualization of Dropping Order for Block Drop and Layer Drop on Larger Models**, i.e., Llama-2-70B and Llama-3-70B.

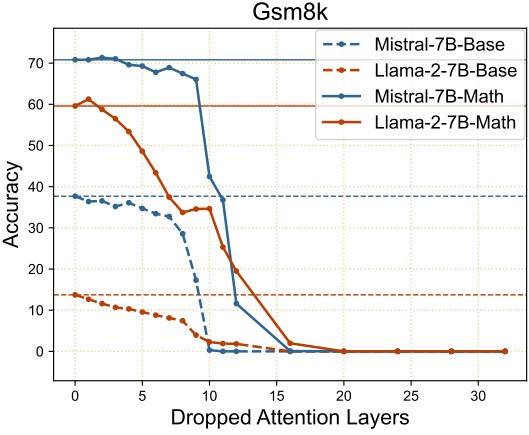

Figure 12: Accuracy Curves on GSM8k.

**Performance on Knowledge Intensive Tasks** To evaluate Attention Drop on more complex technical tasks, we evaluated Llama-2-7B and and Mistral-7B, and two corresponding instruction fine-tuned models, MetaMath-7B-V1.0 and MetaMath-Mistral-7B Yu et al. (2023). The results in Figure 12 indicate that, except for Llama-2-7B-Math, which is MetaMath-7B-V1.0, all the models do not experience significant performance degradation when dropping fewer than 8 Attention layers. We speculate that this is because Llama-2-7B-Math is initialized with Llama-2-7B and undergoes instruction fine-tuning to improve its mathematical ability. Llama-2-7B-Base exhibits poor performance in mathematics, and the ability obtained solely through fine-tuning appears to

be superficial. Therefore, when dropping Attention layers, Llama-2-7B-Math's ability rapidly deteriorates.

**Attention Drop is Orthogonal to Quantization** Given that quantization simplifies data types and enhances efficiency in memory usage and inference speed, we integrate module dropping with quantization to verify whether the quantized models can maintain the performance achieved by Attention Drop. Specifically, we use the mainstream AWQ algorithm Lin et al. (2024) for 4-bit quantization, following its default settings, which involve using 128 samples from the Pile dataset Gao et al. (2020) as the calibration dataset.

As shown in Table 10, the integration of quantization still maintains the performance of Attention Drop, i.e., only less than 1% difference in average performance.

Table 10: **Integration of Module Dropping and Quantization**. "w/Quant" denotes quantized models.

| Method | ARC-C | HellaSwag | OBQA | WinoGrande | Avg. |
|---|---|---|---|---|---|
| Llama-2-13B | | | | | |
| Baseline | 59.9 | 82.2 | 45.6 | 77.0 | 66.2 |
| w/Quant | 59.5 | 81.7 | 45.8 | 77.1 | 66.0 |
| Attn-4 | 58.8 | 82.0 | 46.2 | 77.2 | 66.1 |
| w/Quant | 58.0 | 81.7 | 46.0 | 76.2 | 65.5 |
| Attn-8 | 58.2 | 82.2 | 47.0 | 77.4 | 66.2 |
| w/Quant | 57.7 | 81.9 | 47.0 | 77.0 | 65.9 |
| Mistral-7B | | | | | |
| Baseline | 61.5 | 83.2 | 43.8 | 78.5 | 66.8 |
| w/Quant | 61.2 | 82.5 | 42.8 | 78.0 | 66.1 |
| Attn-4 | 61.0 | 82.9 | 44.6 | 78.0 | 66.6 |
| w/Quant | 61.0 | 82.8 | 43.6 | 77.6 | 66.3 |
| Attn-8 | 60.2 | 82.3 | 44.2 | 78.8 | 66.4 |
| w/Quant | 60.1 | 82.0 | 43.8 | 77.5 | 65.9 |

Table 11: Performance on Long In-Context Task.

| Method | Context Token Length | | | | | Avg. |
|---|---|---|---|---|---|---|
| | 2k | 4k | 7k | 9k | 14k | |
| Baseline | 26 | 70 | 75 | 76 | 81 | 65.6 |
| Attn-2 | 33 | 68 | 71 | 78 | 77 | 65.4 |
| Attn-4 | 28 | 63 | 68 | 75 | 73 | 61.4 |
| Attn-6 | 24 | 63 | 65 | 72 | 69 | 58.6 |
| Attn-8 | 15 | 50 | 53 | 58 | 64 | 48.0 |
| MLP-2 | 29 | 64 | 71 | 70 | 78 | 62.4 |
| MLP-4 | 21 | 57 | 64 | 65 | 66 | 54.6 |
| MLP-6 | 9 | 41 | 48 | 47 | 48 | 38.6 |
| MLP-8 | 11 | 42 | 43 | 48 | 51 | 39.0 |

**Attention Drop on Long In-Context Task** We evaluate the performance of Attention Drop on the long in-context benchmark. Following LongICLBench Li et al. (2024), we present the results on BANKING77 Casanueva et al. (2020), with the only distinction being that we sample 100 examples from the test set. BANKING77 is a banking-domain intent detection dataset comprising 77 classes. We evaluate from 1-shot/label to 5-shot/label, resulting in contextual lengths of 2k, 4k, 7k, 9k, 14k . We employ the togethercomputer/LLaMA-2-7B-32K[2] for Layer Drop, which enlarges the context window of Llama-2 to 32k using position interpolation. From the results in Table 11, we observe that Attention Drop maintains performance and outperforms MLP Drop.

---

[2]https://huggingface.co/togethercomputer/LLaMA-2-7B-32K

