# OpenReview forum: "What Matters in Transformers? Not All Attention is Needed"
_ICLR.cc/2025/Conference — ICLR 2025 Conference Withdrawn Submission_

### Official Review · Reviewer_PbGT · 2024-10-26

**Soundness:** 3
**Presentation:** 3
**Contribution:** 2
**Rating:** 6
**Confidence:** 4

**Summary:**

This paper describes a training-free method for pruning transformers models.

A similarity measure is proposed to estimate the amount of transformation a block (or a layer) apply to their inputs. The similarity measure can be used to define an importance measure which will guide the pruning process.

The importance measure is first applied to prune entire transformer blocks (attention+MLP) from the model. The paper then explains how to apply this measure to estimate the importance of each sublock (attention or MLP) by looking at the residual element within the sublock.
The paper provides empirical block-wise importance data for several popular models, and shows that the earlier and last layer of these models typically exhibit greater importance than the shallow layers.

The paper introduces a Speedup Degradation Ratio metric to help assess the tradeoff between speed and accuracy while pruning a model.

The paper provides KV cache and latency measurements, as a function of how many blocks/layers are pruned.

**Strengths:**

The paper is well written, it is easy to follow and the claims are backed with empirical results, latency measurements and visual insights.

The method is simple and looks easy enough to implement and replicate.

**Weaknesses:**

The novelty of the paper is limited. The paper's main findings (that attention is more easily pruned than MLP, and that shallow layers are more easily pruned than first and last layers) are known, see for example "A deeper look at depth pruning of LLMs" (https://arxiv.org/pdf/2407.16286).

The importance measure is using the scale-invariant cosine similarity measure. It could be argued that this fails to capture magnitude information. Since the cosine similarity measure only depends on the orientation of the vectors, a block that doesn't change the orientation of its input but changes its magnitude could be deemed unimportant. In that case, it might be considered to replace said block with a scaling factor.

There are no comparisons against other methods in the main part of the paper. The appendix has a comparison against ShortenedLLaMA and Wanda. There is no comparison against methods such as LLMPruner, SliceGPT, ShearedLLaMA, or Minitron.

The paper does not show a study on the potential benefits of further fine-tuning after pruning, which could help recover some of the accuracy loss.

The paper does not study how to achieve a finer level of pruning, for example pruning individual attention heads.

The paper does not study the importance of the calibration dataset, or whether domain-specific datasets could be used to improve task-specific benchmarks.

**Questions:**

Can you add a comparison against one or more of LLMPruner, SliceGPT, ShearedLLaMA, or Minitron?

In section 5, it is stated that "For instance, in Llama-2-13B, the KV-cache is reduced from 52GB to 26GB, a 50% reduction. This memory reduction is even more pronounced in larger models like Llama-2-70B, where the KV-cache decreases from 20GB to 10GB.", and the data in the table match this statement. Is this not the other way around, that the 52GB-to-26GB reduction relates to LLama-2-70B?

Did you try to evaluate whether you can prune individual attention heads, as opposed to the whole attention layers?

Did you try to evaluate whether fine-tuning after pruning helps?

Did you try to evaluate whether a scaling factor could be introduced in lieu of a pruned block?

---

> ### Author Response · Authors · 2024-11-16
> **Rebuttal to Reviewer PbGT - Part I**
>
> ## 1. The novelty of our work
>
> [1] is published in July this year, and is a concurrent work of our paper. Other than some similar findings, our work has a lot of key differences with this work:
>
> Our work focuses on exploring **the architecture redundancy of existing in transformers**, for the basic observation of redundancy, redundancy-based dropping techniques, analysis of the effectiveness of attention drop, advanced techniques (Joint Layer Drop) and etc.
>
> Technically, we proposed similarity-based Layer Drop with the consideration of residual connection, performed in an efficient *one-shot* manner, compared to the more computationally expensive Shapley computation in [1]. Additionally, our methods are *training-free* and achieve competitive performance.
>
> Beyond that, we conduct **more thorough experiments in more models and tasks**. Additionally, we comprehensively analyze the efficiency improvements in computation and memory, which provide more informative contributions.
>
> As for the redundancy, **we observe that attention layers exhibit redundancy from the beginning of training and evolve much slower than block or MLP layers**. This finding provides deeper insights into attention architecture design and training strategies.
>
> In summary, while [1] shares similar conclusions, our work is more comprehensive and provides new insights, reinforcing the effectiveness of attention layer redundancy and our dropping techniques.
>
> ### Reference
>
> [1] A deeper look at depth pruning of LLMs, arxiv preprint 2407.16286, 2024
>
> ## 2. Dropping Metrics
>
> In **line 826-842**, we compare our **similarity-based** Layer Drop with **magnitude-based** methods. Magnitude-based dropping techniques like Relative Magnitude work well for small dropping ratios but suffer significant performance degradation at higher ratios. This result showcases the superiority of similarity-based techniques.
>
> Our experiments have already showcased the effectiveness of our methods, and strongly supports our findings. Considering the minimal performance drop between dropped models and the original models, the contribution of more complex methods is trivial.
>
> Consider the residual connection that (y = f(x) + x), where f can be used to represent a block or a layer. Due to the nonlinearity of f, f(x) can be assumed to have deviation with x, and the magnitude of f(x) would also affect the similarity between y and x. However, according to **Figure 3 and 10**, some layers have high similarities that are higher than 0.95 or even 0.99, which cannot be achieved by a high magnitude of f(x).
>
> Thus, although our proposed similarity-based metric is simple, it still implicitly accounts for magnitude.
>
> ## 3. The comparison with other compression techniques.
>
> Our primary contribution lies in discovering the redundancy of attention layers, but we also compare with other compression techniques. Thus, the main part first focuses on the most related contents. Due to the page limit, we leave the comparison with some compression techniques in the appendix, since it does not affect our conclusion.
>
> We have compared our method with several previous works. For instance, we applied the Reverse Order and Relative Magnitude metrics from ShortGPT to Attention Drop as benchmarks, and notably, our Cosine Similarity metric consistently outperformed these alternatives (**see Table 6**).
>
> **In Table 7**, we compare Shortened LLaMA with Joint Layer Drop, where Joint Layer Drop achieves significantly higher speedup while maintaining comparable performance.
>
> We also evaluated our approach against Wanda, an unstructured pruning technique known for its strong pruning performance. Attention Drop shows both better performance and notable speedup, while Wanda does not achieve any speedup.
>
> According to [1], the performance of SliceGPT and LLM-Pruner is significantly weaker than that of ShortGPT. Since Attention Drop has already outperformed both Wanda and ShortGPT, further comparison with LLM-Pruner and SliceGPT becomes less relevant. Additionally, Sheared LLaMA and Minitron are not training-free compression techniques, which differentiates them from the methods under consideration in our work. However, we plan to cite these papers in the related works in the next version.
>
> ### Reference
>
> [1] ShortGPT: Layers in Large Language Models are More Redundant Than You Expect,arXiv preprint arXiv:2403.03853, 2024.

---

> ### Author Response · Authors · 2024-11-16
> **Rebuttal to Reviewer PbGT - Part II**
>
> ## 4. Post-finetuning
> One of the key advantages of our methods is achieving comparable performance and significant efficiency improvements in a **training-free manner**.
> Due to computational constraints, we could not explore post-finetuning, but we acknowledge its potential to restore performance in the Limitations section (**lines 526–531**). This remains a promising avenue for future research, complementing our focus on simple yet effective training-free techniques.
>
> ## 5. Finer level of pruning.
> Our current focus is on layer- and block-level pruning, but we recognize the potential of finer-grained techniques like attention head pruning. Layer Drop and Head Drop are complementary, and their integration could deepen the understanding of attention redundancy within the AI community. Our goal is to explore the impact of varying degrees of pruning across different layers. If attention layers are redundant, it follows naturally that attention heads within them may also exhibit redundancy. While some recent works after the ICLR submission dealine have begun to explore this [1], it does not diminish the novelty of our approach.
> [1] DuoAttention: Efficient Long-Context LLM Inference with Retrieval and Streaming Heads
>
> ## 6. The effect of calibration datasets
> In **lines 799-805**, we systematically discuss the effect of calibration datasets, where we conduct ablation studies with different datasets and sample sizes. **Figures 2, 3, and 10** demonstrate that the importance scores are robust to changes in the calibration datasets, both in terms of dataset type and sample size. As a result, the dropped layers or blocks remain largely unaffected by variations in the calibration data.
> The phenomenon of varying levels of performance degradation across different tasks is commonly observed in model compression studies, though performance can often be restored through post-finetuning. However, due to limited computational resources, we were unable to perform this step and instead highlight the potential of post-training in the Limitations section (**lines 526-531**).

---

> ### Author Response · Authors · 2024-11-16
> **Rebuttal to Reviewer PbGT - Part III**
>
> ## Q1. Comparison with other compression techniques
> We have compared our methods with ShortGPT, Shortened LLaMA and Wana, and the results are presented in the Appendix.
>
> ## Q2. Clarification of KV-Cache.
> Thanks for this question. There is no misalignment between models and memories. The KV-Cache of Llama-2-13B is high, since it does not deploy Group-Query Attention. Thanks for pointing this out, we will refine the sentence in the next version.
>
> ## Q3. Effect of further finetuning.
> Thanks for your suggestion. The effect of further finetuning after pruning is admitted. Since our work focuses on simple but effective training-free dropping techniques and the limited computation resources, we did not afford training the model and list post-finetuning in the Limitation.
>
> ## Q4. Finer Levels Pruning Like Attention Heads.
> Our work supports the effectiveness of dropping attention heads. And there is some work like DuoAttention began to do this. Layer Drop and Head Drop are complementary to each other, and we believe their integration enhances the understanding of redundancy of attention within the Ai community.
>
> ## Q5. Scaling Factor in a pruned block.
> Thanks for your suggestions. From previous works and our ablation study, we found the magnitude seems not as effective as similarity-based techniques.
> While the proposed similarity-based dropping techniques have already demonstrated competitive performance and support our claims, we will continue exploring more advanced techniques to further enhance performance.

---

> ### Author Response · Authors · 2024-11-20
> **Thanks**
>
> We truly appreciate your recognition of the relevance of training-free optimization and the value of our experimental results and ablation studies. Your improved rating and constructive comments motivate us to continue refining and advancing this line of research. Thank you again for your review and support.

---

### Official Review · Reviewer_rs71 · 2024-11-03

**Soundness:** 3
**Presentation:** 2
**Contribution:** 2
**Rating:** 5
**Confidence:** 4

**Summary:**

- investigates the redundancy within different modules of Transformer-based large language models, including Blocks, MLP, and Attention layers.
- The redundancy is evaluated using a similarity-based metric.
- This paper validates that some model components can be pruned with obvious speedup and minor performance drop.

**Strengths:**

- The paper is clearly written and well organized, it provides a systematic exploration of redundancy in Transformers, focusing on Blocks, MLP, and Attention layers.
- This paper provides several useful insights, for example:
	- FFNs seem more important and Attention modules can be dropped with minimal performance impact with high efficiency
	- deeper layers seem less important compared t the shallower ones, which indicated the model has obtained anwsers in early layers.
- The findings in this paper have practical implications for deploying LLMs more efficiently in real-world applications by reducing deployment costs and resource demands.
- The experiments are extensive and the efficiency of the method is validated to be consistent on different tasks and models.

**Weaknesses:**

- All experiments in this paper are conducted on a group of datasets, however these datasets are still limited and cannot represent the real-world applications and validate the generalization ability of the pruned model. For example, if the input sequence is not short, early layer attention modules can model the token-wise relationships and predict correct anwsers, but long sequence tasks such as needle in a haystack might be seriously affected by the dropping.
- In addition, the importance scores rely on calibration datasets, and the paper does not extensively explore how variations in these datasets might affect the pruning results. More details should be given, why performances degradation on some datasets are extensive but ignorable on others.

**Questions:**

I wonder the results on more complex tasks and long sequence tasks such as longbench and needle in a haystack.

---

> ### Author Response · Authors · 2024-11-16
> **Rebuttal to Reviewer rs71**
>
> ## 1. Long context and complex tasks
>
> Thank you for your question regarding the performance of our method on long-context and complex tasks. We present the experimental results for a **long-context task** in **lines 945-966**, where the Attention Drop consistently outperforms the MLP Drop, underscoring the effectiveness of our approach in handling extended contexts while maintaining computational efficiency. Additionally, in **lines 895-920**, we show results for **knowledge-intensive tasks** like GSM8K, where the models maintain comparable performance even after dropping some attention layers. These findings further support the redundancy of attention layers. While these tasks are inherently more demanding, we note that post-training has the potential to recover any performance loss caused by the dropping process [3,4], making it a promising avenue for future exploration.
>
>
> ## 2. Robustness to calibration datasets
>
> **In lines 799-805**, we systematically discuss the effect of calibration datasets, where we conduct ablation studies with different datasets and sample sizes. **Figures 2, 3, and 10** demonstrate that the importance scores are robust to changes in the calibration datasets, both in terms of dataset type and sample size. The dropped layers or blocks remain largely unaffected by these variations, reinforcing the flexibility and generalizability of our method.
>
> ## 3. Different Performance Changes
>
> As observed in other model compression studies[1,2,3,4], performance degradation across different tasks is a common phenomenon, but such degradation can often be mitigated through post-finetuning. However,  we were unable to perform post-finetuning for our experiments. Nonetheless, we emphasize its potential to restore performance in the Limitations section (**lines 526–531**). This remains an important area for future research and practical application.
>
> ### References
> [1] The Unreasonable Ineffectiveness of the Deeper Layers, arXiv 2024.
>
> [2] ShortGPT: Layers in Large Language Models are More Redundant Than You Expect, arXiv 2024.
>
> [3] Shortened LLaMA: Depth Pruning for Large Language Models with Comparison of Retraining Methods, arXiv 2024.
>
> [4] A Simple and Effective Pruning Approach for Large Language Models, 	ICLR 2024.

---

> ### Author Response · Authors · 2024-11-21
> **Follow Up Reminder I**
>
> Dear Reviewer rs71,
>
> As we have responded for serval days, we would like to cordially inquire about the extent to which we have successfully addressed the concerns outlined in your review.
>
> Should any lingering points require further attention, please rest assured that we are enthusiastic about the opportunity to provide comprehensive responses to any subsequent queries or comments you may have.
>
> Your constructive input remains invaluable to us, and we appreciate your dedication to enhancing the quality of our manuscript. Thank you for your time and consideration.
>
> Best,
>
> Submission8274 Authors

---

> ### Author Response · Authors · 2024-11-23
> **Follow Up Reminder II**
>
> Dear Reviewer rs71,
>
> As we near the end of the rebuttal period, we would like to kindly inquire whether we have adequately addressed the concerns outlined in your review.
>
> If there are any remaining points requiring further clarification, please rest assured that we are fully committed to providing detailed responses to any additional queries or comments you may have.
>
> Your constructive feedback has been invaluable, and we deeply appreciate your effort in helping to improve the quality of our manuscript.
>
> Thank you for your time and thoughtful consideration.
>
> Best regards,
> Submission8274 Authors

---

> ### Author Response · Authors · 2024-11-25
> **Follow Up Reminder III**
>
> Dear Reviewer rs71,
>
> As the rebuttal period draws to a close, we apologize for the repeated reminder; we simply want to ensure that we have fully addressed all of your concerns.
>
> If there are any remaining points that require further clarification, please rest assured that we are committed to providing detailed responses to any additional queries or comments you may have.
>
> Your constructive feedback has been invaluable, and we deeply appreciate the time and effort you've dedicated to improving the quality of our manuscript.
>
> Thank you once again for your thoughtful consideration.
>
> Best regards,
> Submission8274 Authors

---

> > ### Author Response · Authors · 2024-11-27
> > **Follow Up Reminder IV**
> >
> > Dear Reviewer rs71,
> >
> > We apologize for the repeated reminder. It has been 10 days since we submitted our responses, but we have not yet received your feedback. **We simply want to ensure that we have fully addressed all your concerns**.
> >
> > In your initial review, you acknowledged our contributions to addressing layer redundancy. Your primary concerns, however, focused on long-context tasks and the impact of different calibration datasets. **We have provided detailed responses to these points in our reply to you and in the general responses**. We sincerely hope that our efforts address your concerns effectively.
> >
> > If there are any remaining points that require further clarification, please rest assured that we are fully committed to providing detailed responses to any additional queries or comments you may have.
> >
> > **As the discussion period has been extended, we are eager to make full use of this time to engage in meaningful dialogue and address any lingering concerns**. Your constructive feedback has been invaluable, and we deeply appreciate the time and effort you have dedicated to improving the quality of our manuscript.
> >
> > Thank you once again for your thoughtful consideration.
> >
> > Best regards,
> > Submission8274 Authors

---

> ### Author Response · Authors · 2024-11-30
> **Follow Up Reminder V**
>
> Dear Reviewer rs71,
>
> We hope this message finds you well. Apologies for the repeated reminder, but it has been over 10 days since we submitted our responses, and we have not yet received your feedback.
>
> We have provided detailed responses in general responses and responses to you  and would greatly value your feedback on whether they address your concerns.
>
> If further clarification is needed, we remain fully committed to addressing any additional queries.
>
> With the discussion period extended, we are eager to engage further and sincerely appreciate your constructive feedback to improve our manuscript.
>
> Thank you for your time and consideration.
>
> Best regards,
> Submission8274 Authors

---

> > ### Author Response · Authors · 2024-12-03
> > **Follow Up Reminder VI**
> >
> > Dear Reviewer rs71,
> >
> > We hope this message finds you well. Apologies for the urgency, but as today marks the final day of the discussion period, we are reaching out again regarding your feedback.
> >
> > It has been over 10 days since we submitted our responses, and we have not yet heard back from you. We provided detailed replies to your comments and would sincerely appreciate knowing if they have adequately addressed your concerns.
> >
> > If there are any remaining questions or points of clarification, we remain fully committed to addressing them promptly before the discussion period concludes.
> >
> > Your feedback is invaluable for ensuring a fair and thorough evaluation of our work, and we deeply appreciate your time and effort.
> >
> > Thank you for your consideration.
> >
> > Best regards,
> > Authors of Submission8274

---

### Official Review · Reviewer_gGuV · 2024-11-04

**Soundness:** 2
**Presentation:** 2
**Contribution:** 2
**Rating:** 5
**Confidence:** 5

**Summary:**

This paper research into the redundancy of attention and MLP layers of transformer-based LLMs. The paper proposes a layer pruning methods with similarity-based layer redundancy measurement. By tracing model checkpoints throughout the training process, it is shown that the layer redundancy is inherent and consistent across training stages. The experiments demonstrate that the pruning method speedup the methods while preserve the performance to some extend.

**Strengths:**

1. The paper proposed an detailed method for Attention and MLP pruning.
2. The paper is well-written and easy to understand.

**Weaknesses:**

1. The comtribution of this paper is limited. There are many previous similar methods in layer pruning [1][2][3], and this paper is a simple extension to the MLP and attention layers.
2. The author did not analyze the reason for the layer redundancy. Although extensive experiments are provided, the reason why transformer-based LLMs exhibit redundency on the MLP and Attention layers are not explained.
3. The author did not compare the experimental performance with previous block or layer pruning methods [1][2][3].


[1] Song J, Oh K, Kim T, et al. SLEB: Streamlining LLMs through Redundancy Verification and Elimination of Transformer Blocks[J]. arXiv preprint arXiv:2402.09025, 2024.

[2] Men X, Xu M, Zhang Q, et al. Shortgpt: Layers in large language models are more redundant than you expect[J]. arXiv preprint arXiv:2403.03853, 2024.

[3] Zhang Y, Li Y, Wang X, et al. FinerCut: Finer-grained Interpretable Layer Pruning for Large Language Models[J]. arXiv preprint arXiv:2405.18218, 2024.

**Questions:**

1. Why a normally trained Transformer model exhibits redundancy in layer levels? Since there are no explicit inductive bias of layers, it is more likely that each layer learns distinct information for the final prediction. Why pruning several layers do not affect the performance?

---

> ### Author Response · Authors · 2024-11-16
> **Rebuttal to Reviewer gGuV - Part I**
>
> ## 1. Difference with Previous Works
>
> Our approach addresses several gaps and limitations in previous works on layer pruning, offering distinct advantages in efficiency, analysis, and applicability:
>
> (1) **Scope of Pruning**:
>    Previous works [1,2] primarily focus on block-level pruning and largely ignore **internal architectures like MLPs and Attention layers**. Block-level pruning is also insufficient to maintain the performance of original models.While FinerCut [3] applies layer pruning, our method is significantly more efficient with a computational complexity of \( O(L) \) compared to their \( O(L^2) \).
>
> (2) **Efficiency and Performance Improvements**:
> Our method is simple but effective and demonstrates the significant improvement of attention dropping than previous block drop techniques. In addition, our method is designed in a training-free manner, making it especially suitable for resource-constrained scenarios while achieving competitive performance.
>
> (3) **Residual Connections**:
>    We have also demonstrated the **unique context of Layer Drop**, specifically addressing the role of residual connections in enabling effective attention layer dropping. This is well-supported by our ablation study on residual connections, as shown in **lines 794-798 and Table 7**, which highlights their critical role in the effectiveness of our method.
>
> (4) **Comprehensive Analysis**:
> We provide comprehensive experimental results on **how the dropping techniques enhance the efficiency in memory and computation**, while previous works mainly focus on performance itself. **We also provide our observation of importance scores and their dynamics during training in line 443-475**, which demonstrates the property of attention architectures and provides informative insight for further model design and training.
>
> ### References
> [1] Song J, Oh K, Kim T, et al. *SLEB: Streamlining LLMs through Redundancy Verification and Elimination of Transformer Blocks*. arXiv preprint arXiv:2402.09025, 2024.
> [2] Men X, Xu M, Zhang Q, et al. *ShortGPT: Layers in large language models are more redundant than you expect*. arXiv preprint arXiv:2403.03853, 2024.
>
> ---
>
> ## 2. The Reason Why Attention Layers Are Redundant
>
> Regarding this question, our primary observation is based on the similarity between hidden states before and after the attention layers. The results indicate that attention layers do not significantly transform the hidden states. This aligns with previous work [1], which suggests that token similarity increases substantially in shallow layers and quickly reaches saturation. Given the high similarity between tokens, the attention mechanism, which aggregates these tokens, does not cause a major transformation of the hidden states.
>
> We have also demonstrated the dynamics of redundancy in attention layers during training in **Figure 7**. Specifically, we observe that the importance scores of attention layers start very low and evolve much more slowly compared to MLP layers and Blocks. This is attributed to both the inherent redundancy in attention mechanisms and the limitations of current training strategies in mitigating this redundancy.
>
> ### Reference
> [1] Revisiting Over-Smoothing in BERT from the Perspective of Graph, ICLR 2022.

---

> ### Author Response · Authors · 2024-11-16
> **Rebuttal to Reviewer gGuV - Part II**
>
> ## 3. Comparison with Previous Pruning Methods
>
> We have conducted extensive comparisons with existing pruning techniques to highlight the strengths of our approach. For instance, we applied the Reverse Order and Relative Magnitude metrics from ShortGPT to Attention Drop as benchmarks, and notably, our Cosine Similarity metric consistently outperformed these alternatives (see **Table 6**).
>
> In **Table 7**, we compare Shortened LLaMA with Joint Layer Drop, where Joint Layer Drop achieves significantly higher speedup while maintaining comparable performance.
>
> We also evaluated our approach against Wanda, an unstructured pruning technique known for its strong pruning performance. Attention Drop shows both better performance and notable speedup, while Wanda does not achieve any speedup.
>
> Among the mentioned papers, SLEB primarily focuses on Block Drop rather than Attention Drop, with performance deteriorating significantly as more layers are dropped. ShortGPT is included for comparison in Table 6. FINERCUT has a complexity of O(L^2), whereas our method operates with a more efficient complexity of O(L). Furthermore, since FINERCUT is still under review and not yet open-sourced, we plan to include it in our comparisons as soon as its code becomes available.
>
> ## 4. Why Transformers Exhibit Redundancy in Layer Levels
>
> The redundancy in Transformer layers arises from the residual connection architecture. Each Transformer layer incorporates a residual connection, i.e., y = f(x) + x. When f(x) is weak, y is dominated by x, resulting in high similarity between the input and output.
>
> This high similarity means that certain layers do not significantly alter the hidden states, making them effectively redundant. Consequently, pruning these layers has minimal impact on model performance.

---

> ### Author Response · Authors · 2024-11-21
> **Follow Up Reminder I**
>
> Dear Reviewer gGuV,
>
> As we have responded for serval days, we would like to cordially inquire about the extent to which we have successfully addressed the concerns outlined in your review.
>
> Should any lingering points require further attention, please rest assured that we are enthusiastic about the opportunity to provide comprehensive responses to any subsequent queries or comments you may have.
>
> Your constructive input remains invaluable to us, and we appreciate your dedication to enhancing the quality of our manuscript. Thank you for your time and consideration.
>
> Best,
>
> Submission8274 Authors

---

> ### Author Response · Authors · 2024-11-23
> **Follow Up Reminder II**
>
> Dear Reviewer gGuV,
>
> As we near the end of the rebuttal period, we would like to kindly inquire whether we have adequately addressed the concerns outlined in your review.
>
> If there are any remaining points requiring further clarification, please rest assured that we are fully committed to providing detailed responses to any additional queries or comments you may have.
>
> Your constructive feedback has been invaluable, and we deeply appreciate your effort in helping to improve the quality of our manuscript.
>
> Thank you for your time and thoughtful consideration.
>
> Best regards,
> Submission8274 Authors

---

> ### Author Response · Authors · 2024-11-25
> **Follow Up Reminder III**
>
> Dear Reviewer gGuV,
>
> As the rebuttal period draws to a close, we apologize for the repeated reminder; we simply want to ensure that we have fully addressed all of your concerns.
>
> If there are any remaining points that require further clarification, please rest assured that we are committed to providing detailed responses to any additional queries or comments you may have.
>
> Your constructive feedback has been invaluable, and we deeply appreciate the time and effort you've dedicated to improving the quality of our manuscript.
>
> Thank you once again for your thoughtful consideration.
>
> Best regards,
> Submission8274 Authors

---

> ### Author Response · Authors · 2024-11-27
> **Follow Up Reminder IV**
>
> Dear Reviewer gGuV,
>
> We apologize for the repeated reminder. It has been 10 days since we submitted our responses, but we have not yet received your feedback. **We simply want to ensure that we have fully addressed all your concerns**.
>
> Your main concerns appear to relate to our technical contributions and the underlying reasons for layer redundancy. **We have provided detailed responses to these questions in both our reply to you and the general response**. May we kindly ask if you could spare some time to review our responses and share your feedback?
>
> If there are any remaining points that require further clarification, please rest assured that we are committed to providing detailed answers to any additional queries or comments you may have.
>
> **Since the discussion period has been extended, we are eager to fully utilize this time to engage in meaningful discussions with you and address your concerns**. Your constructive feedback has been invaluable, and we deeply appreciate the effort and time you've dedicated to improving the quality of our manuscript.
>
> Thank you once again for your thoughtful consideration.
>
> Best regards,
> Submission8274 Authors

---

> > ### Author Response · Authors · 2024-11-30
> > **Follow Up Reminder V**
> >
> > Dear Reviewer gGuV,
> >
> > We hope this message finds you well. Apologies for the repeated reminder, but it has been over 10 days since we submitted our responses, and we have not yet received your feedback.
> >
> > Your concerns regarding our technical contributions and layer redundancy have been addressed in detail in our responses. Could you kindly review them and share your thoughts?
> >
> > If further clarification is needed, we are committed to addressing any remaining questions promptly.
> >
> > With the discussion period extended, we are eager to engage further and greatly value your feedback in improving our manuscript.
> >
> > Thank you for your time and consideration.
> >
> > Best regards,
> > Submission8274 Authors

---

> ### Comment · Reviewer_gGuV · 2024-11-30
> **Post-rebuttal discussion**
>
> Thanks for the detailed responses. After carefully reading the rebuttal, I believe the study on layer redundancy is important. However, the author do not explain why the MLP layer can also be pruned. I think that the novelty of layer pruning is somewhat limited comparing with the previous layer pruning methods (e.g. ShortGPT) and more fundamental analysis is needed. Thus, I choose to keep my initial rating.

---

> ### Author Response · Authors · 2024-11-30
> **Request Reviewer gGuV for Clarification on the Feedback**
>
> Dear Reviewer gGuV,
>
> Thank you for your reply. However, we are somewhat confused by your comments and would appreciate clarification.
>
> **While MLP Drop serves as a baseline in our work, it is not the focus of our study**. Furthermore, **we do not advocate for pruning MLP layers**. **Could you elaborate on why an explanation for pruning MLP layers is deemed necessary?** We believe this aspect is not central to our work and does not significantly impact our contributions.
>
> Regarding ShortGPT, as noted in our results, it employs pruning at the block level. **Our findings clearly show that Block Drop is suboptimal compared to Attention Drop**, supporting our approach. We have also explicitly explained the specificity and significance of Attention Drop in both our paper and rebuttals.
>
> As for the analysis, we have already visualized the importance scores for Attention, MLP, and Block layers. **These visualizations demonstrate that MLP and Block layers are considerably more critical than Attention layers, further underscoring our methodological choices**.  We also provided our responses in both the orignal paper and previous responses about redundancy at layer level.  While some MLP layers were dropped for Joint Layer Drop, we have explicitly explained that only a small proportion of MLP layers exhibit low importance, as stated in lines 468–469.
>
>
> Your feedback has left us uncertain about the specific concerns raised. We kindly request that you reconsider your points and provide clarification during the remaining days of the review process.
> ##  **If you choose to maintain your statements, we kindly ask you to provide sufficient evidence that directly aligns with our work**.
>
> Your insights are valuable, and we want to ensure that we have fully addressed your concerns.
> Thank you once again for your time and consideration.
>
> Best regards,
> Submission 8274 Authors

---

### Official Review · Reviewer_PgkH · 2024-11-04

**Soundness:** 2
**Presentation:** 3
**Contribution:** 3
**Rating:** 6
**Confidence:** 4

**Summary:**

This paper studies the redundancy across different modules within Transformers,
including Blocks, MLP, and Attention layers. The study shows that the redundancy mainly comes from the Attention. The authors thus propose an “Attention Drop” method for parameter pruning in LLMs. Experiments on Llama and Mistral are conducted to demonstrate the effectiveness of the proposed method.

**Strengths:**

- The paper is well-organized and easy to follow.
- The study on Attention redundancy is interesting. The proposed approach is reasonable.

**Weaknesses:**

-	The results show that Attention layers and last layers have more redundancy. However, the experiments are mainly conducted on Llama and Mistral. Is this finding valid for other LLMs? It would be interesting to show more LLMs.
-	Please further clarify how the dropped layers are selected in the pruning process. If the target is to drop 4 layers, the attention layers are tested one by one to find 4 layers with lowest importance score?

**Questions:**

Please see above.

---

> ### Author Response · Authors · 2024-11-16
>
> ## 1. Applicability to Other LLMs
>
> Thank you for your suggestion regarding the applicability of our method to other LLMs. Our proposed attention drop method is inherently **model- and task-agnostic**, making it versatile across a range of architectures. Beyond Mistral and LLaMA, we have tested our method on additional models. For example, in **Table 8**, we compare Shortened LLaMA with Joint Layer Drop on **Vicuna-13B-v1.3**, where Joint Layer Drop achieves significantly higher speedup while maintaining comparable performance.
>
> Additionally, we **provide more results of Yi-9B, Solar-10.7b, Baichuan-2-7b, and DeepSeek-MoE-16B** here:
>
> ### DeepSeek-MoE-16B Results
>
> | Method      | Drop_n | HellaSwag | MMLU  | OBQA  | Winogrande | Avg.   |
> |-------------|--------|-----------|-------|-------|------------|--------|
> | Baseline    |    –   | 80.0      | 44.5  | 43.6  | 73.8       | 60.5   |
> | MLP Drop    | 5      | 69.8      | 38.2  | 38.0  | 70.4       | 54.1   |
> |             | 10     | 39.4      | 31.4  | 32.6  | 53.9       | 39.3   |
> | Attn Drop   | 5      | 79.6      | 44.2  | 45.8  | 74.6       | 61.1   |
> |             | 10     | 74.0      | 38.5  | 39.4  | 72.8       | 56.2   |
>
> ### Yi-9B Results
> | Method      | Drop_n | HellaSwag | MMLU | OBQA | Winogrande | Avg.  |
> |-------------|--------|-----------|------|------|------------|-------|
> | Baseline    |    –   | 80.4      | 69.6 | 45.0 | 79.0       | 68.5  |
> | MLP Drop    | 4      | 75.1      | 67.2 | 38.6 | 74.0       | 63.7  |
> |             | 8      | 67.4      | 63.7 | 29.2 | 71.7       | 58.0  |
> |             | 12     | 63.4      | 60.2 | 28.6 | 69.9       | 55.6  |
> | Attn Drop   | 4      | 79.9      | 69.6 | 44.0 | 79.2       | 68.2  |
> |             | 8      | 79.1      | 68.2 | 42.4 | 78.2       | 67.0  |
> |             | 12     | 78.0      | 66.3 | 42.0 | 77.0       | 65.8  |
>
> ### Solar-10.7b Results
> | Method      | Drop_n | HellaSwag | MMLU | OBQA | Winogrande | Avg.  |
> |-------------|--------|-----------|------|------|------------|-------|
> | Baseline    |    –   | 84.6      | 64.0 | 43.4 | 82.6       | 68.7  |
> | MLP Drop    | 4      | 84.0      | 63.5 | 43.6 | 81.9       | 68.3  |
> |             | 8      | 79.0      | 63.1 | 40.2 | 80.3       | 65.6  |
> |             | 12     | 73.3      | 61.5 | 36.2 | 76.6       | 61.9  |
> |             | 16     | 67.5      | 57.4 | 33.8 | 72.5       | 57.8  |
> | Attn Drop   | 4      | 84.4      | 64.1 | 43.8 | 82.2       | 68.6  |
> |             | 8      | 84.4      | 64.1 | 44.0 | 83.0       | 68.9  |
> |             | 12     | 83.8      | 63.6 | 44.0 | 82.0       | 68.3  |
> |             | 16     | 82.3      | 63.5 | 43.2 | 81.5       | 67.6  |
>
> ### Baichuan-2-7b Results
>
> | Method      | Drop_n | HellaSwag | MMLU | OBQA | Winogrande | Avg.  |
> |-------------|--------|-----------|------|------|------------|-------|
> | Baseline    |    –   | 73.8      | 54.0 | 39.6 | 70.7       | 59.5  |
> | MLP Drop    | 4      | 67.7      | 44.7 | 36.4 | 69.1       | 54.5  |
> |             | 8      | 54.2      | 36.8 | 33.6 | 66.6       | 47.8  |
> |             | 12     | 44.2      | 47.9 | 28.2 | 62.0       | 45.6  |
> | Attn Drop   | 4      | 74.1      | 54.1 | 39.8 | 71.3       | 59.8  |
> |             | 8      | 73.4      | 53.8 | 40.8 | 71.8       | 60.0  |
> |             | 12     | 65.1      | 44.9 | 37.4 | 69.7       | 54.3  |
>
> The performance and efficiency **remain consistent with the results presented in our paper**. The additional models include not only mainstream dense large language models but also Mixture of Experts models, further demonstrating the **versatility and applicability** of our methods across various Transformer architectures.
>
> We appreciate your feedback, and while we have already evaluated our method on mainstream LLMs, we plan to extend this work to additional architectures in future research to further validate its generalizability.

---

> ### Author Response · Authors · 2024-11-16
> **Rebuttal to Reviewer PgkH - Part II**
>
> ## 2. How to Drop Models by More than One Layer
>
> Thank you for your question regarding the process of dropping more than one layer.
> As explained in **line 222**, our Layer Drop approach operates in a **one-shot** manner. This means that we compute the importance scores for all layers once and remove redundant layers in a single step.
>
> To address your point about **iterative layer dropping**, we also include an ablation study comparing one-shot dropping to iterative dropping, where the latter, as you mentioned, removes redundant layers one by one. The results, presented in **Figure 9**, show that **iterative dropping achieves performance only comparable to one-shot dropping**, without delivering significant improvements. Considering the simplicity and computational efficiency of one-shot dropping, we have prioritized it as the default approach in our work.
>
> To improve clarity in future versions, we will include additional annotations and explanations to ensure this distinction is easily understood.
>
> Thank you again for your thoughtful feedback and suggestions. Your insights are invaluable in helping us refine our work further.

---

> ### Author Response · Authors · 2024-11-21
> **Follow Up Reminder I**
>
> Dear Reviewer PgkH,
>
> As we have responded for serval days, we would like to cordially inquire about the extent to which we have successfully addressed the concerns outlined in your review.
>
> Should any lingering points require further attention, please rest assured that we are enthusiastic about the opportunity to provide comprehensive responses to any subsequent queries or comments you may have.
>
> Your constructive input remains invaluable to us, and we appreciate your dedication to enhancing the quality of our manuscript. Thank you for your time and consideration.
>
> Best,
>
> Submission8274 Authors

---

> ### Author Response · Authors · 2024-11-23
> **Follow Up Reminder II**
>
> Dear Reviewer PgkH,
>
> As we near the end of the rebuttal period, we would like to kindly inquire whether we have adequately addressed the concerns outlined in your review.
>
> If there are any remaining points requiring further clarification, please rest assured that we are fully committed to providing detailed responses to any additional queries or comments you may have.
>
> Your constructive feedback has been invaluable, and we deeply appreciate your effort in helping to improve the quality of our manuscript.
>
> Thank you for your time and thoughtful consideration.
>
> Best regards,
> Submission8274 Authors

---

> ### Author Response · Authors · 2024-11-25
> **Follow Up Reminder III**
>
> Dear Reviewer PgkH,
>
> As the rebuttal period comes to a close, we would like to confirm if we have sufficiently addressed your concerns, which primarily focused on applicability and algorithmic details.
>
> To address applicability, we performed extensive experiments on both dense and MoE models, which we believe respond to your comments. Additionally, we provided an explanation on how to handle cases where multiple layers are dropped, and we plan to further refine this explanation in the next version of the manuscript.
>
> Did these efforts resolve your concerns? If there are any points that still require clarification, please let us know, and we will gladly provide additional details.
>
> Your thoughtful feedback has been invaluable in improving our work, and we sincerely appreciate your time and effort.
>
> Best regards,
> Submission8274 Authors

---

> ### Author Response · Authors · 2024-11-25
> **Request for Clarification on Regarding Novelty of Layer Pruning**
>
> Thank you for your feedback. However, we would like to seek clarification on certain points.
>
> Could you please clarify your concerns regarding the novelty of our work and layer pruning? In your initial review, **you acknowledged our contribution in proposing the method**, but we would like to emphasize that our work is the first to systematically study this issue and identify attention redundancy as a key challenge.
>
> Moreover, **our proposed Layer Dropping approach is simple and effective**, and we believe that simplicity and effectiveness are strengths that make a method **widely applicable and impactful**, rather than detracting from its novelty. If you maintain that **layer pruning** lacks novelty, could you kindly provide specific points?
>
> Best regards,
>
> Submission8274 Authors

---

### Author Response · Authors · 2024-11-16
**General Response to Reviewers**

Dear Reviewers,

Thank you for your time and effort in reviewing our paper. We appreciate your feedback and are grateful for the opportunity to refine our work further. **We have noticed some recurring questions and concerns, and we would like to address them collectively here before providing detailed responses to individual reviews**.

**Due to space limitations, we had to move certain content to the Appendix**, including comparisons with other compression techniques, ablation studies on calibration datasets, and experiments on knowledge-intensive and long-context tasks, among others. **We kindly request that you revisit these sections in the Appendix, as they directly address some of the concerns raised**. It appears that certain claims regarding weaknesses may stem from these sections being overlooked during the review process.

In addition to the proposed techniques, **we focus on the observation and analysis of the redundancy of attention layers in Section 6, which represents one of our key contributions**. This analysis forms the foundation of our proposed techniques and provides key insights into their effectiveness. **We invite you to revisit this section, as it may help clarify several points raised in your feedback**.

Once again, we sincerely thank you for your constructive feedback and the time you have invested in evaluating our submission. We hope these clarifications provide a better context for understanding our work. Your insights are invaluable, and we greatly appreciate your assistance in refining our paper further.

Sincerely,
Authors of Submission 8274

---

### Author Response · Authors · 2024-11-26
**General Response on Clarifying the Novelty of Our Paper**

Dear Reviewers,

Thank you for the time and effort each of you has dedicated to reviewing our submission. We appreciate your insights, but we believe there are some misunderstandings regarding our contributions that merit clarification.

First, while prior works have identified layer redundancy, **many of them are either concurrent or predate the era of large language models. These works do not diminish our contributions.** Instead, we offer a comprehensive investigation into the redundancy of attention layers, including how to represent it, address it, and explore potential underlying reasons. **Our contribution is both insightful and holistic**, rather than narrow or incremental.

Second, our work specifically focuses on **architectural redundancy through the lens of layer/block analysis**, utilizing layer drop and block drop as the most straightforward techniques to address this issue. **These techniques offer greater efficiency** compared to finer-granularity pruning approaches. Therefore, **claiming that layer pruning lacks novelty is unwarranted**.

**Last but not least, while our methods are simple, they are highly effective, and their novelty should not be underestimated.** We have highlighted the unique challenges posed by layer drop as distinct from block drop. Importantly, our approach maintains a complexity of O(L) compared to the O(L^2) or higher complexity of other methods. Experimental results have already validated the effectiveness of our techniques, and **introducing unnecessary complexity on algorithms would detract from their elegance and efficiency**.

We sincerely appreciate your time and effort in reviewing our work and respectfully request that you reconsider your evaluation. Should you have further questions or require additional clarification, please do not hesitate to reach out.

Best regards,
Submission8274 Authors

---

### Public Comment · ~Weilin_Cai1 · 2024-12-05

I find it very interesting that there are many attention modules in the transformer model that can be dropped.
This approach is simple yet effective, offering valuable insights for my exploration of more efficient transformer model structures.
Additionally, the code provided in the Supplementary Material is very user-friendly and indeed reproduces the optimization results consistent with those described in the paper.

---

### Note · Authors · 2025-01-23

I have read and agree with the venue's withdrawal policy on behalf of myself and my co-authors.